# Plastic vasomotion entrainment

**Daichi Sasaki, Ken Imai, Yoko Ikoma, Ko Matsui***

Super-network Brain Physiology, Graduate School of Life Sciences, Tohoku University, Sendai, Japan

**Abstract** The presence of global synchronization of vasomotion induced by oscillating visual stimuli was identified in the mouse brain. Endogenous autofluorescence was used and the vessel 'shadow' was quantified to evaluate the magnitude of the frequency-locked vasomotion. This method allows vasomotion to be easily quantified in non-transgenic wild-type mice using either the wide-field macro-zoom microscopy or the deep-brain fiber photometry methods. Vertical stripes horizontally oscillating at a low temporal frequency (0.25 Hz) were presented to the awake mouse, and oscillatory vasomotion locked to the temporal frequency of the visual stimulation was induced not only in the primary visual cortex but across a wide surface area of the cortex and the cerebellum. The visually induced vasomotion adapted to a wide range of stimulation parameters. Repeated trials of the visual stimulus presentations resulted in the plastic entrainment of vasomotion. Horizontally oscillating visual stimulus is known to induce horizontal optokinetic response (HOKR). The amplitude of the eye movement is known to increase with repeated training sessions, and the flocculus region of the cerebellum is known to be essential for this learning to occur. Here, we show a strong correlation between the average HOKR performance gain and the vasomotion entrainment magnitude in the cerebellar flocculus. Therefore, the plasticity of vasomotion and neuronal circuits appeared to occur in parallel. Efficient energy delivery by the entrained vasomotion may contribute to meeting the energy demand for increased coordinated neuronal activity and the subsequent neuronal circuit reorganization.

*For correspondence:
matsui@med.tohoku.ac.jp

**Competing interest:** The authors declare that no competing interests exist.

## eLife assessment

This article presents **important** results indicating a plastic enhancement in the vasomotion response of pial cortical arterioles to external stimulation in awake mice using a wide range of external visual stimulation paradigms. The evidence for this interesting effect, with broad potential applications, is **solid**. These results are relevant for scientists and clinicians interested in the regulation of blood flow in the brain.

## Introduction

Energy consumption in the brain is considered to be super-efficient compared to that of modern computers (*Attwell and Laughlin, 2001*; *Howarth et al., 2012*; *Sokoloff, 1960*). Revealing the principle underlying the relationship between energy delivery, metabolism, and information processing would lead to our understanding of the unique feature of the biological brain that evolved to optimally utilize the limited energy supply. The neuronal circuit requires an adequate energy supply for the proper functioning of information encoding and inducing plastic changes. Energy (i.e., oxygen and glucose) is delivered to the brain via the blood vessels. The on-demand vascular motion appears to allow localized energy delivery to specific brain regions or global regulation of the efficiency of energy delivery.

Vasodilation and vasoconstriction spontaneously occurring at seconds time scale (~0.1 Hz) have been reported (*Davis et al., 2008*; *Fujii et al., 1990*; *Hundley et al., 1988*; *Jones, 1853*). Such spontaneous 'vasomotion' without any apparent external stimuli was first described in the vein of the bat

wing (*Jones, 1853*) and was subsequently observed mainly in the arteries and arterioles of numerous species (*Fujii et al., 1990*; *Hundley et al., 1988*). In mice, the temporal pattern of such spontaneous vasomotion is two orders of magnitude different from the frequency of heartbeat (~10 Hz; *Ho et al., 2011*; *Kramer et al., 1993*). Therefore, vasomotion should be considered distinct from pulsation due to heartbeat. The slow vasomotion is likely beneficial for the efficient delivery of blood in small vessels as has been indicated by mathematical modeling (*Meyer et al., 2002*). Vasomotion has also been suggested to facilitate the clearance of unwanted substances such as Aβ (*Aldea et al., 2019*; *van Veluw et al., 2020*).

Although vasomotion can spontaneously occur on its own without the presentation of any apparent external stimuli, it can also be affected by neurovascular coupling mechanisms (*Rivadulla et al., 2011*; *van Veluw et al., 2020*; *Villringer and Dirnagl, 1995*). Neuronal firing triggers the release of vasodilator signals from neurons and/or astrocytes (*Zhu et al., 2022*). In response to these signals, the blood vessels dilate through the relaxation of the smooth muscle cells (SMCs) and the pericytes (*Davis et al., 2008*; *Peppiatt et al., 2006*). Due to these signals, the frequency, amplitude, or phase of an already ongoing spontaneous vasomotion can be affected. Whether the blood vessel is spontaneously oscillating or static, it can react to the external presentation of sensory stimuli. The vasodilation response to the neural activity caused by the sensory stimulation is termed 'functional hyperemia' (*Nippert et al., 2018*; *Stickland et al., 2019*; *Zhu et al., 2022*). Functional hyperemia is triggered by the neurovascular coupling mechanisms. The functional MRI signals largely reflect signals created by the changes in the blood flow (*Glover, 2011*; *Ogawa et al., 1990*).

Here, both spontaneous vasomotion and functional hyperemia induced by horizontally oscillating visual stimuli at temporal frequencies similar to the naturally occurring spontaneous vasomotion were studied. In this article, the term 'vasomotion' is used to describe both forms of vascular motion. Spontaneous vasomotion and visually induced vasomotion were examined on the surface of the cerebral cortex through intact crania using a macro-zoom microscope or studied with an optical fiber inserted deep in the cerebellum using fiber photometry methods in vivo mice.

We show the global entrainment of the visually induced vasomotion throughout the surface of the brain in mice with repeated visual stimulation. Functional hyperemia confined to a sensory-specific region has been previously reported (*van Veluw et al., 2020*). Our unique observation was that the visually induced vasomotion developed with repeated training and became perfectly frequency-locked to the presented stimuli. Such plastic entrainment occurred globally throughout the whole cortex, and it was not limited to the visual cortex.

The horizontally oscillating visual pattern that was used in this study is known to induce optokinetic response (HOKR). Repeated exposure to such a pattern results in an increased amplitude of the HOKR eye movement (*Ito, 1982*; *Kanaya et al., 2023*; *Waespe and Henn, 1977*). Neuronal circuit plasticity required for the HOKR learning takes place in the flocculus region of the cerebellum. Visually induced vasomotion dynamics at the flocculus, the deep brain region, were studied using the fiber photometry method. A concomitant increase in the HOKR performance and the magnitude of the frequency-locked vasomotion was observed. These results suggest the plastic change of vasomotion dynamics may support and interact with the neuronal circuit plasticity.

## Results

### Presence of spontaneous vasomotion

Blood vessels in the brain can change their diameter through the actions of the SMCs and the pericytes (*Davis et al., 2008*; *Peppiatt et al., 2006*). Here, we observed spontaneous vasomotion in vivo in unanesthetized head-restrained awake mice. We sought to understand how vascular motion is regulated by sensory stimuli, how visually induced vasomotion can be entrained, and the functional significance of vasomotion plasticity in information processing in the brain.

Anesthesia or the creation of an invasive cranial window could potentially perturb naturally occurring vasomotion. Therefore, we studied the presence of spontaneous vasomotion in the brain of awake wild-type mice through the skull using a macro-zoom microscope. To visualize the blood vessels, a fluorescence dye, TexasRed dextran, was administrated through the tail vein. Blood vessels on the surface of the brain could be visualized through an intact skull when the skull is coated with a transparent resin; however, the images were inevitably blurred. Therefore, in initial experiments,

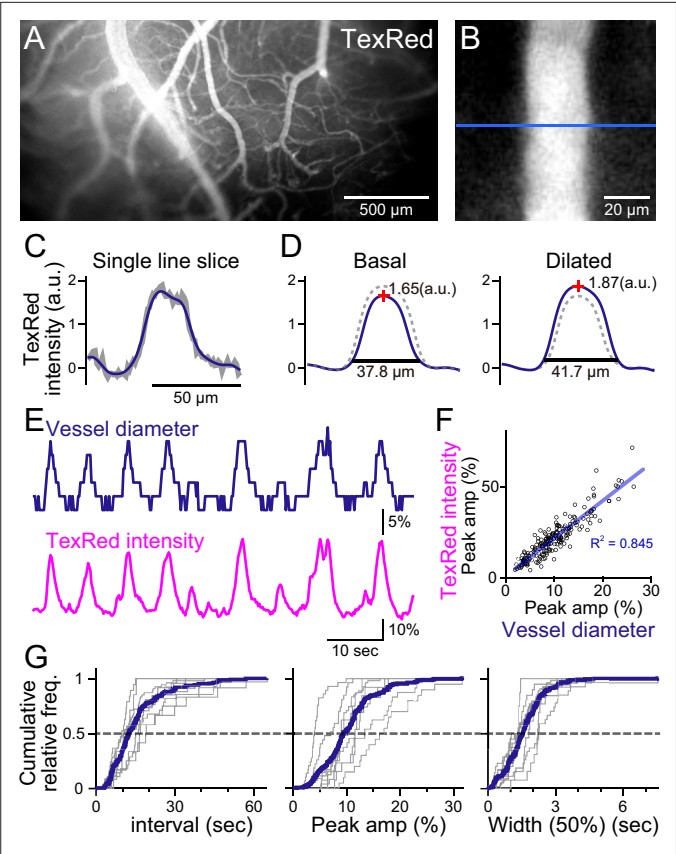

**Figure 1.** Spontaneous vasomotion detection. (**A**) A fluorescence image of a mouse brain through the thinned-skull cranial window using a single-photon wide-field macro-zoom microscope. TexasRed was intravenously injected and the blood vessels on the brain surface (between bregma and lambda) could be readily resolved. (**B**) Magnified image of a single blood vessel. TexasRed fluorescence intensity profile was calculated at the cross-section indicated by the blue line. (**C**) An example of a single-line intensity profile. Raw traces (gray bold line) were filtered to derive the boundaries between the vessels and parenchyma. (**D**) TexasRed intensity profiles averaged for multiple lines. The profile during the basal, undilated phase (solid line shown on the left, the same profile shown as a dotted line on the right) and the dilated phase (solid line on the right, dotted line on the left). Peak amplitudes of the intensity profiles (red cross) and full width at 10% maximum (horizontal lines). (**E**) An example of a time-series data of the calculated vessel diameter and TexasRed intensity. (**F**) Correlation of the peak amplitude of the calculated vessel diameter with the TexasRed peak intensity (n = 214 peaks). (**G**) Characteristics of the vessel dilation events across multiple vessels were summarized. Inter-event interval (left), the peak amplitude of the vessel diameter (middle), and the half-width of the duration of individual events (right) are shown for individual sessions collected from multiple vessels (gray lines, n = 9 sessions from five vessels of three animals; duration of each session was from 1.5 to 11 min). Averaged cumulative histograms are shown as blue lines.

the thinned-skull cranial window (*Drew et al., 2010*; *Morii et al., 1986*) was created (*Figure 1A*). Spontaneous vasodilation/vasoconstriction oscillations were observed. To quantify the spontaneous vasomotion dynamics, we attempted to measure the vessel diameter; however, even through the thinned-skull cranial window, the exact boundary between the vessel and the brain parenchyma was difficult to determine. Therefore, the TexasRed fluorescence intensity profile perpendicular to the axis of blood flow in the vessel was measured, and the full width at 10% maximum of the TexasRed fluorescence was taken as an indication of vessel diameter (*Figure 1B and C*; 45.7 ± 9.6 [S.D.] μm, n = 5 vessels observations; see validation of the use of full width at 10% maximum for vessel diameter estimation in the 'Materials and methods' section). The diameter of the selected vessels suggests that mostly the pial arteries or the arterioles were observed for the stereomicroscope studies.

Spontaneous vessel diameter dynamics could be revealed; however, the quantification was pixelated due to the resolution limit of the microscope (*Figure 1E*, top). Therefore, the peak TexasRed fluorescence within the blood vessels was also quantified. The vessel diameter and the peak TexasRed

fluorescence correlated well; that is, a larger vessel diameter resulted in a higher peak fluorescence (*Figure 1F*). Higher resolution of the vasomotion dynamics could be examined with the peak fluorescence measurement method (*Figure 1E*, bottom). Spontaneous vasodilation events were observed with the near-instantaneous rise and an apparent exponential decay. The cumulative frequency of the inter-event interval, the peak amplitude of the dilated vessel diameter, and the half-width of the duration of individual events were summarized (*Figure 1G*). The vasodilation events occurred relatively periodically (15.8 ± 4.8 [S.D.] s), were relatively fixed in amplitude (10.4 ± 3.7 [S.D.]%), and the half-width of duration was also relatively constant (1.6 ± 0.4 [S.D.] s; n = 9 sessions from five vessels observations).

It is possible that even the creation of the thinned skull may perturb the naturally occurring vasomotion as heat or vibration created by the drilling was reported to influence the surface of brain tissue (*Augustinaite and Kuhn, 2020*). Thus, the blood vessels were observed through the intact skull coated with a transparent resin. The blood vessels could be surprisingly well resolved with this method without the use of a two-photon microscope; however, compared to the thin skull observation, the boundary between the vessel and the brain parenchyma was more vague (*Figure 2A–C*). TexasRed fluorescence intensity profile perpendicular to the vessel was often shallower than that observed with the thinned skull; however, the full width at 10% maximum of the profile and the peak intensity appeared to be correlated in the intact skull as well (*Figure 2D*). As the intensity profile was shallow with intact skull imaging, the full width at 10% maximum of the profile appeared to be less reliable as an indicator of vessel diameter and measurements of the TexasRed intensity fluctuations would better represent the vessel volume changes stemming from vasomotion.

During these macro-zoom microscopy recordings, we noticed that an intense autofluorescence from the brain tissue could be detected with ~430 nm excitation and ~540 nm emission. Most of the autofluorescence appeared to be coming from the brain parenchyma. The inside of the blood vessels was mostly devoid of this autofluorescence. Therefore, with the simultaneous TexasRed and autofluorescence imaging, the blood vessels and the brain parenchyma could be separately visualized (*Figure 2E and F*). Dynamic fluorescence intensity changes of the TexasRed and autofluorescence were simultaneously observed (*Figure 2G*). The two fluorescence traces nearly completely mirrored each other (*Figure 2H and I*). Therefore, the inverted autofluorescence dynamics would also reflect the vasomotion dynamics. In later experiments, 400-µm-diameter optical fibers will be used to measure the vasomotion dynamics in the deep brain areas. To simulate the situation using images from the brain surface, a 500-µm-diameter circular region of interest (ROI) was drawn and the average TexasRed fluorescence dynamics were plotted (*Figure 2J and K*). As apparent from the spectrogram of the trace, several cycles of oscillatory fluctuations at low frequencies (0.1–0.5 Hz) were spontaneously observed. The frequency of the oscillatory fluctuation within a cluster was not strictly fixed at a constant value. The occurrence of clusters was also not periodic, and the duration of a cluster also varied. The presence of similar low-frequency vascular oscillations has been reported in previous studies (*Davis et al., 2008*; *Jones, 1853*), but in those studies, the cycle of fluctuation appeared to be constant and the periodic cycles continued without much interruption. It is possible that spontaneous vasomotion detected in our system through the intact skull in awake in vivo mice was less periodic because spontaneous behaviors such as whisking, twitching, or struggling could cause functional hyperemia resulting in the interruption of apparent periodic oscillations.

## Entrainment of vasomotion with visual stimuli presentation

Horizontally oscillating visual stimuli are known to induce involuntary eye movement and the line of sight follows the stimuli in head-fixed mice. This response to the stimuli is termed horizontal optokinetic response (HOKR). Repeated presentation of the stimuli also induces a gradual increase in the amplitude of the eye movement and the small flocculus region of the cerebellum has been shown to be responsible for this learning (*Aziz et al., 2014*; *Ito, 1982*). Whether visually induced vasomotion (or functional hyperemia) occurs in response to this HOKR-inducing visual stimuli in the cortex was examined. Immediately after TexasRed was tail vein injected, using the whole brain surface imaging with the macro-zoom microscope, TexasRed fluorescence fluctuation in response to the HOKR stimuli was studied (*Figure 3A and B*).

Horizontally oscillating vertical black and white stripes were presented to the head-fixed mouse with a temporal frequency of 0.25 Hz and with a maximal visual angle (amplitude) of 17° for 15 min

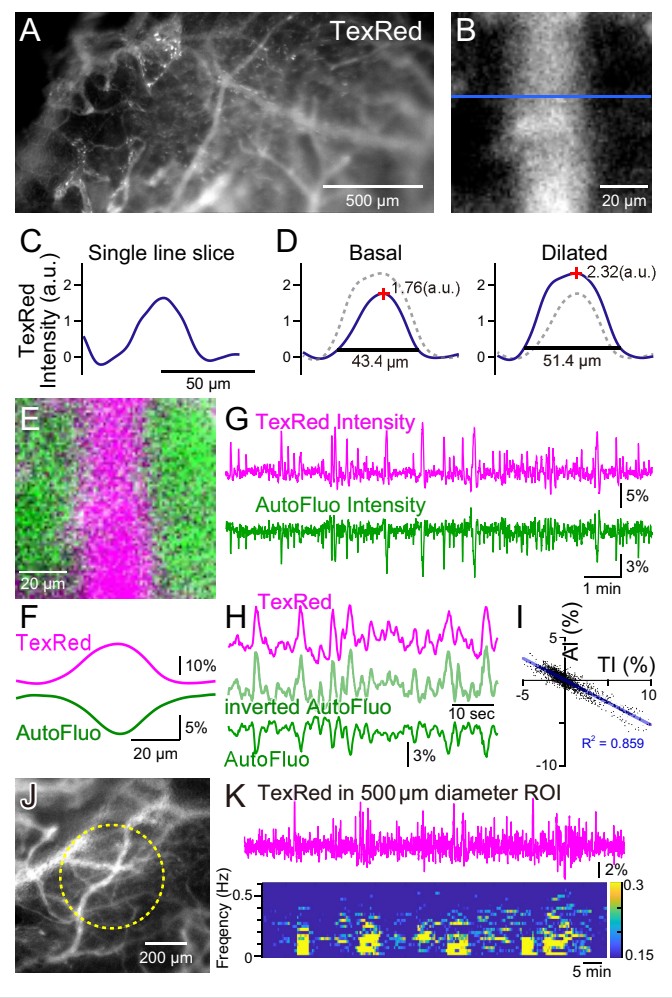

**Figure 2.** Autofluorescence signals could be used for vasomotion detection through the intact skull. (**A**) A fluorescence image of a mouse brain through the intact skull (between bregma and lambda). Although with less resolution, individual blood vessels containing TexasRed could be visualized. (**B**) Magnified image of a single blood vessel. (**C**) A single-line intensity profile. (**D**) Averaged TexasRed intensity profile at the basal (left) and the dilated (right) states. The intensity profiles were shallower compared to the profiles analyzed with thinned-skull images. Therefore, rather than the vessel diameter estimated by the full width at 10% maximum of the intensity profile, the TexasRed intensity would likely better reflect the volume changes of the blood vessels. (**E**) A merged image of the TexasRed signal in the blood vessel (magenta) and the autofluorescence signal likely coming exclusively from the brain parenchyma (green). (**F**) Spatial intensity profile along the cross-section of TexasRed and autofluorescence signals in (**E**). (**G**) The temporal transitions of the TexasRed and the autofluorescence intensity signals at a region of interest (ROI) encompassing the vessel are shown in (**E**). The two signals mostly mirrored each other, suggesting that vessel dilation appears as the TexasRed intensity increase and the autofluorescence intensity decrease. (**H**) An enlarged segment of the traces is shown in (**G**). The inverted autofluorescence trace (light green) resembles the TexasRed intensity trace. (**I**) Autofluorescence intensity (AI) plotted against the TexasRed intensity (TI) for the duration of the traces shown in (**G**) (n = 3440 data points). (**J**) An image of the mouse brain through the intact skull. 500 diameter μm circular ROI is shown (dotted yellow line). This diameter corresponds closely to the diameter (400 μm) of the optical fiber used in fiber photometry experiments. (**K**) Spontaneous TexasRed fluctuation (top) was detected through the intact skull within the ROI shown in (**J**). The spectrum below shows that the low temporal frequency vasomotion does not continuously occur and short phases of vascular oscillation occur with frequency not locked to a narrow specific range.

per session (*Figure 3A*). In response to the first visual training session, fixed frequency oscillation of the TexasRed intensity at the primary visual cortex (V1) was not observed. TexasRed signals in these 'Novice' mice, fluctuated with non-fixed low frequencies which consisted mainly of <0.3 Hz (*Figure 3C*, left). After a couple of more training sessions and just before the fourth training session,

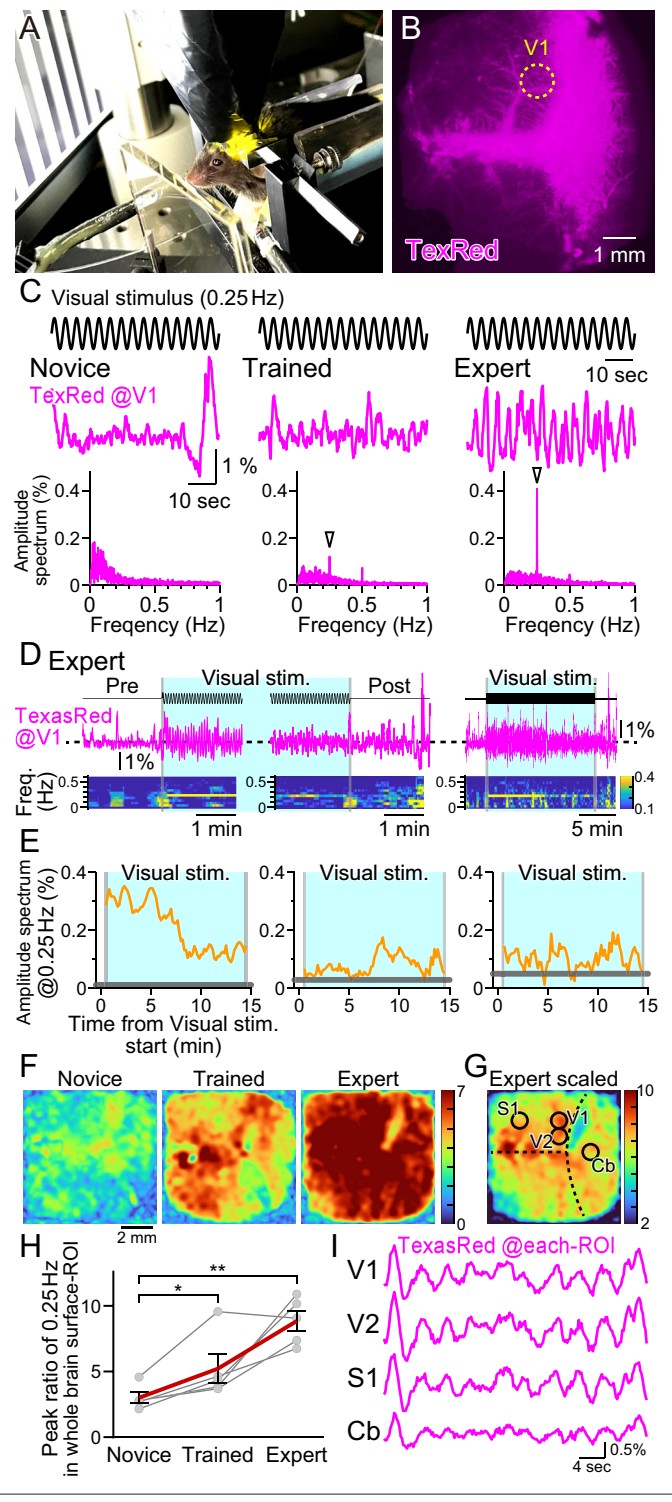

**Figure 3.** Vasomotion entrainment with the oscillating visual stimulus. (**A**) Fluorescence images were taken from the whole brain surface through the intact skull using the macro-zoom microscope in awake mice. Horizontally oscillating (at 0.25 Hz temporal frequency) vertical stripes are shown as the visual stimuli on the two perpendicular monitors placed in front of the mouse. (**B**) Representative image of the near-whole brain surface after intravenous injection of TexasRed. The location of region of interest (ROI) in V1 is shown with the yellow dotted line. (**C**) TexasRed signals at V1-ROI during visual stimulation are shown. After repeated sessions, the vasomotion gradually became entrained to the frequency of the visual stimulus. At the first training session (Novice, left), no

*Figure 3 continued on next page*

*Figure 3 continued*

specific peak was observed in the amplitude spectrum (bottom) of the TexasRed signals (middle). After several trainings (Trained, at fourth training session, middle), a 0.25 Hz peak (arrowhead) in the amplitude spectrum appeared. After saturated training (Expert, at 8th–11th training session, right), the 0.25 Hz oscillation of the TexasRed signal was apparent in the raw traces (middle). (**D**) An example data from an Expert mouse. Immediately after the onset of a visual stimulation session, the 0.25 Hz band appeared in the TexasRed trace (left). The 0.25 Hz band dissipated immediately upon the cessation of the visual stimulation (middle). The 0.25 Hz band lasted throughout the 15 minute session in this particular mouse, albeit the amplitude became smaller toward the end of the session (right). (**E**) The amplitude values at 0.25 Hz in the spectrogram were plotted against time for three different animals. The left graph is from the same Expert animal and the same session as shown in (**D**). The two other graphs (middle, right) are from different sessions. The amplitudes at 0.25 Hz during the visual stimulation were mostly higher than those before the stimulation (horizontal gray line); however, the amplitude did not stay constant in most cases. The vasomotion appears to come in and out of the frequency lock to the visual stimulus. (**F**) The peak amplitude at 0.25 Hz over the mean amplitude in the 0.1–1 Hz range (PR0.25) was calculated for each pixel in the whole brain during the visual stimulation for Novice, Trained, and Expert mice. PR0.25 increase with repeated sessions was not restricted to the visual cortex but a near-uniform increase was observed throughout the whole brain surface. The color bar indicates PR0.25. (**G**) The color scale for the Expert example shown in (**F**) is saturated. Therefore, a different color scale (right) was applied for the same data. (**H**) The averaged PR0.25 in the whole brain surface ROI gradually increased as the training progressed (n = 5 animals). Novice vs. Trained, p=0.039, *p<0.05; Novice vs. Expert, p=0.0030, **p<0.005; paired *t*-test with Holm correction for multiple test. Data from individual animals are shown in gray, and the averaged data across all animals are shown in red. Data are presented as mean ± SEM, as error bars. (**I**) TexasRed signals in ROIs at V1, V2, S1, and cerebellum (Cb). Signals in different locations fluctuated with similar frequency with similar phases.

---

TexasRed was administered again. In these 'Trained' mice, the TexasRed signal in the cortex fluctuated periodically. Fourier transform of the TexasRed signal at V1 during the 15 min training session revealed a clear sharp peak in the amplitude spectrum at 0.25 Hz, which was not observed in the Novice mice (*Figure 3C*, middle). This 0.25 Hz is the temporal frequency of the HOKR oscillatory visual stimuli. We also noticed that, although the amplitudes were often much smaller, second and third harmonic signals were sometimes observed. The second harmonic signal could be explained if visually induced vasomotion reacts to both directions of oscillating stimuli. The origin of the third harmonics was undetermined. These harmonic signals were not always observed, and the magnitude of these signals was variable compared to the robust frequency-locked signal.

With further repeated training sessions (8–11 training sessions), the TexasRed signal fluctuation further becomes more strongly tuned to the frequency of the visual stimuli ('Expert', *Figure 3C*, right). In this Expert mouse, the 0.25 Hz fluctuation in the TexasRed signal becomes visible even in the raw traces. Using this Expert mouse, the time course of the vasomotion frequency locked to the visual stimulus was examined (*Figure 3D*). Before the visual stimulus, the amplitude at 0.25 Hz was negligible; however, as soon as the visual stimulus started, a sharp rise in 0.25 Hz frequency amplitude increased. The amplitude gradually declined but stayed significant until the cessation of the visual stimulus and the vasomotion amplitude locked to the visual stimulus frequency quickly dissipated (*Figure 3D and E*, left panel). This characteristic of an instant onset and a steady increase was not always shared in all animals. In many cases, the frequency locked to the 0.25 Hz came in and out throughout the visual stimulation session; however, mostly throughout the session, the vasomotion amplitude at 0.25 Hz frequency was above the baseline measured before the visual stimulation (examples in *Figure 3E*, middle and right panel).

Visual stimulus affecting vasomotion or hyperemia in the V1 was more or less expected; however, does it affect other regions of the cortex? To examine this, data from the near-whole brain surface observation with a macro-zoom microscope was studied. As the TexasRed fluorescence was low, acquired images were Gaussian filtered in the spatial XY dimension to reduce the pixelated noise at the expense of spatial resolution reduction without losing much of the temporal resolution. Then, the TexasRed signal fluctuation during the 15 min session was examined for every pixel imaged. The peak of the amplitude of the TexasRed signal at 0.25 Hz temporal frequency was calculated and was divided with the mean amplitude at 0.1–1 Hz range (peak ratio of 0.25 Hz; PR0.25). This normalization process was necessary because each region had a difference in the baseline fluctuation level of the fluorescence signal. Color-coded map of the PR0.25 was mostly uniform throughout the cortex and gradually increased from Novice, Trained, to Expert (*Figure 3F*). Averaged PR0.25 in the whole brain

surface ROI gradually increased as the training progressed (*Figure 3H*). The increase in the PR0.25 was so great in the Expert that the color map was mostly saturated in *Figure 3F*, right panel. Therefore, the color code was scaled down in *Figure 3G*. The location of V1, secondary visual cortex (V2), primary sensory cortex (S1), and the cerebellum (Cb) was roughly identified based on the coordinates of the mouse brain atlas by referring to the bregma and lambda locations. PR0.25 in V1 and V2 was high but not particularly high compared to S1 or Cb or, in fact, anywhere else that could be observed. An example of the actual TexasRed signal fluctuation trace in various ROI is shown in *Figure 3I*. The 0.25 Hz signal fluctuations were apparent in all ROI and, in addition, the phase of the signal fluctuation was also nearly synchronized (n = 5 animals; data not shown). These data show that frequency and phase-locked vasomotion are not restricted to the visual-related area, and enhancement and entrainment of vasomotion are observed widely throughout the surface of the brain.

## The use of autofluorescence signal for examination of vasomotion

The small cerebellar flocculus region deep and away from the cranial surface is responsible for the cerebellar-dependent HOKR motor learning. To examine the vasomotion in such a deep brain region, optical fiber was implanted into the cerebellum (*Figure 4C*). The eye movement amplitude normally increases in response to repeated sessions of HOKR visual stimulation training. However, we found that, if TexasRed had to be injected before every training session, the mouse did not learn very well. It is possible that the stress induced by the tail vein injection interfered with the learning process. However, as demonstrated below, an already acquired HOKR learning with multiple training sessions does not get lost with the stressful injection. An alternative method to measure the vessel volume dynamics without the use of TexasRed fluorescence in the blood vessels was required to examine the development of visually induced vasomotion during the training sessions. To accomplish this, the use of fluorescence in the brain parenchyma was attempted.

As shown in *Figure 4A*, the blood vessels are devoid of the fluorescence from the YFP expressed in astrocytes, and thus the blood vessels appear as dark regions in the images. Therefore, dilation or constriction would appear as a decrease and increase in the fluorescence signal of the YFP. Such concept of 'shadow' imaging of the dynamics of brain blood volume has been demonstrated using the fiber photometry method previously (*Ikoma et al., 2023a*; *Ikoma et al., 2023b*). Here, we first used a transgenic mouse, *Mlc1-tTA::tetO-YC$_{nano50}$*, which expresses FRET-based CFP and YFP conjugated sensor protein specifically in astrocytes, including Bergmann glial cells (*Kanemaru et al., 2014*; *Tanaka et al., 2012*). Instead of exciting the CFP with ~430 nm, YFP was directly excited by delivering ~505 nm light through the implanted optical fiber in the cerebellum, and the YFP fluorescence (dYFP) was detected using the same optical fiber. The excitation filter used strictly restricted the exposed light from exciting the CFP (see 'Materials and methods' for details). The dYFP fluorescence would not be affected by the Ca$^{2+}$ and would only be affected by the local brain blood volume and possibly the cytosolic pH as YFP protein is sensitive to changes in the pH. It is known that YFP fluorescence can be quenched by acidification. Both TexasRed and dYFP signals simultaneously monitored with fiber photometry fluctuated with the 0.25 Hz temporal frequency directly matching the visual stimulus, but mirrored in the direction of change (*Figure 4D*). The TexasRed signal oscillations were locked to the temporal frequency of the visual stimulus. This shows that, not only in the cortex but also deep in the cerebellum, vasomotion can be induced with visual stimulation. Cross-correlation of the TexasRed and dYFP signals shows a negative peak with ~0 s lag time (Δt; *Figure 4E and F*). The lag time (Δt) between TexasRed and dYFP was calculated and the phase shift was determined by dividing Δt by the presented visual stimulation oscillation cycle time (5 or 4 s, depending on the experiment), which was then multiplied by 360° and was added 180°. The average phase shift of dYFP relative to TexasRed was 184.4° ± 17.6° (S.D.) (*Figure 4G*; n = 13 segments with 5 min durations each), indicating that these signals were nearly completely inverted. This shows that dYFP signals can be used for assessing the vasomotion dynamics. The cytosolic pH may change also; however, even if there are effects of the cytosolic pH fluctuations on the fluorescence amplitude, pH changes appear to be in phase with the oscillatory vasomotion.

Unfortunately, robust HOKR learning could not be reliably induced in this line of transgenic mice for unknown reasons. Therefore, C57BL/6NJcl wild-type mice were used for the experiments described in the rest of the manuscript. As introduced in *Figure 2E*, autofluorescence excited with 430 nm light could be detected in the wild-type mice. Using the images taken with the macro-zoom microscope

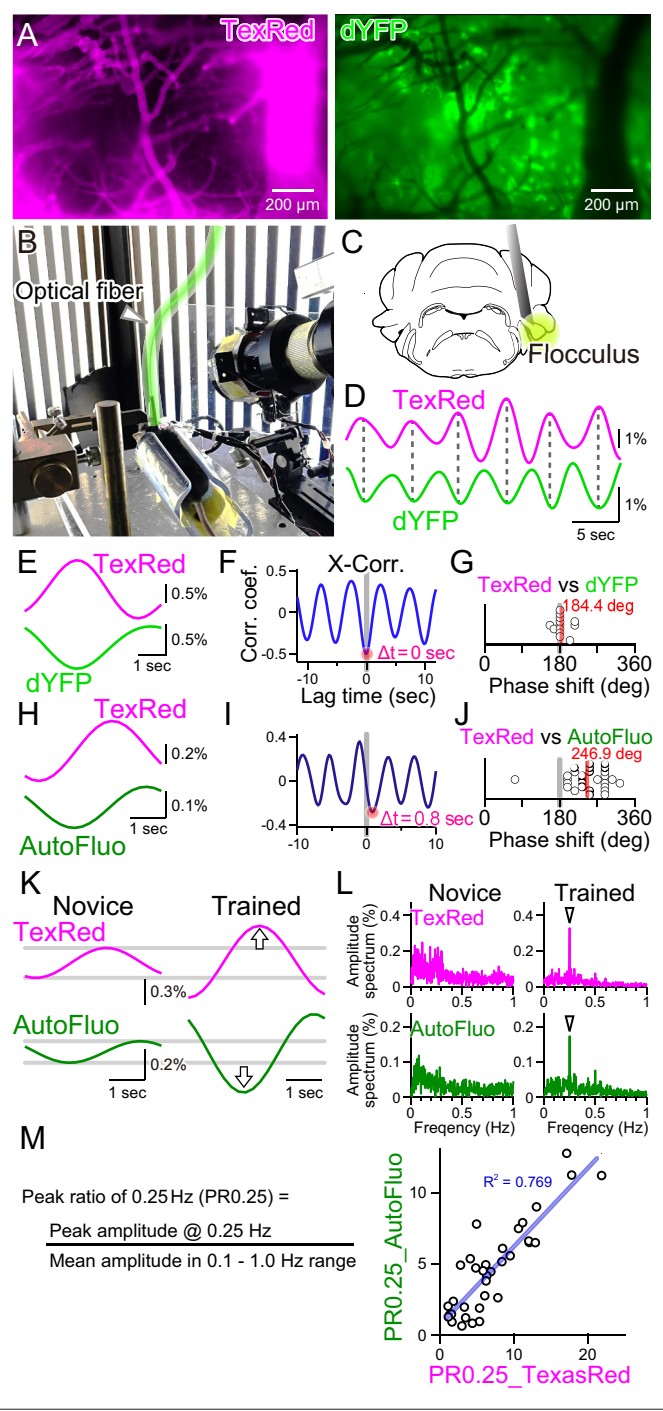

**Figure 4.** Autofluorescence signals could be used to quantify the vasomotion entrainment. (**A**) Images of the TexasRed fluorescence in the blood vessels and the direct YFP fluorescence expressed in astrocytes of a transgenic mouse. The blood vessels appear as dark 'shadows' in the dYFP fluorescence image. (**B**) An optical fiber was implanted in the cerebellum close to the flocculus and another optical fiber (highlighted in green) from the fiber photometry apparatus was connected to this implanted fiber. Horizontally oscillating visual stimulation was presented on the monitors. A hot mirror reflected the IR image of the eye and the eye location was monitored using a camera. (**C**) Location of the optical fiber implantation relative to the cerebellar flocculus. (**D**) Simultaneously recording of the TexasRed and the dYFP signals from the optical fiber implanted in the flocculus during the horizontal optokinetic response (HOKR) visual stimulation session. The dYFP mirrored the TexasRed signal fluctuations. Fluorescence signal data were filtered at 150–350 mHz. (**E**) 20 5 s segments of the TexasRed and

*Figure 4 continued on next page*

*Figure 4 continued*

the dYFP signal traces were extracted and averaged. The two traces nearly completely mirrored each other. The temporal frequency of the visual stimulation presented to the transgenic mouse was 0.20 Hz. (**F**) Cross-correlation between the two traces had a negative peak with a lag time of Δt of 0 s. (**G**) The phase shift between TexasRed and dYFP signals was calculated for n = 13 segments with 5 min durations each. (**H**) In a separate set of experiments, the TexasRed signal and the endogenous autofluorescence signal were simultaneously recorded in wild-type mice. 20 4 s segments were extracted and averaged. The traces were not completely inverted and a certain amount of phase shift was observed. The temporal frequency of the visual stimulation presented to the wild-type mouse was 0.25 Hz. (**I**) Cross-correlation of TexasRed and Autofluorescence signals. Δt of 0.8 s was observed in this particular example. (**J**) Phase shifts in TexasRed and autofluorescence signals were calculated for n = 28 segments. (**K**) Averaged data of 35 traces of the extracted 4 s segments of the TexasRed and the autofluorescence signals during the first training (Novice) and the fourth training (Trained). Data was filtered with 150–350 mHz. An increase in the amplitudes was observed from both signals. (**L**) Amplitude spectrums for TexasRed and autofluorescence during the 5 min of the first training (Novice) and that of the fourth training (Trained). Signal data were high-passed filtered at 50 mHz. (**M**) PR0.25 of the TexasRed signals and the autofluorescence signals in the 5 min during the visual stimulation were plotted against each other. A clear correlation was observed indicating that the autofluorescence signal can be used to estimate the magnitude of the vasomotion frequency-locked to the visual stimulus.

and limiting the ROI in a small region surrounding the blood vessels, mirrored traces of the TexasRed and autofluorescence were observed (***Figure 2H***). However, when the signals collected using the 400 μm diameter fiber photometry were analyzed, TexasRed fluorescence and autofluorescence did not mirror completely and a significant shift in the phase was observed (***Figure 4H–J***; 246.9° ± 49.9° [S.D.], n = 28 segments). The endogenous substance in the brain parenchyma that produces autofluorescence was not identified but it could be flavin or other related substances (***Croce and Bottiroli, 2015***; ***Huang et al., 2002***; ***Islam et al., 2013***). Production and degradation of flavin and other metabolites may be induced by the fluctuation in the blood vessel diameter with a fixed delay time. The phase shift in the autofluorescence could be due to the additive effect of 'shadow' imaging of the vessel and to the concentration fluctuation of the autofluorescent metabolite.

The TexasRed fluorescence and the autofluorescence signals are shown for the initial sessions (Novice, ***Figure 4K***) and after several training sessions (Trained). The amplitude of the 0.25 Hz temporal frequency component of both the TexasRed and the autofluorescence signals increased with training (***Figure 4K and L***). PR0.25 of autofluorescence was plotted against that of the TexasRed for multiple training sessions and a strong correlation between these two parameters was found (***Figure 4M***). Therefore, although the autofluorescence dynamics result from the fluctuations reflecting the 'shadow' of vasomotion and an unknown fluctuation source, the PR0.25 apparently reflects the amplitude of the vasomotion locked to the visual stimulation cycle. In the remainder of the article, the autofluorescence fluctuations were mainly examined to understand the dynamics of the visually induced vasomotion.

## Vasomotion followed the visual stimulation dynamics over a wide parameter range

Optimal visual stimulation properties for inducing frequency-locked vasomotion were searched. The visual stimulation used in all other sections of this study was the following. The temporal frequency of horizontal oscillation was 0.25 Hz; the spatial amplitude of the horizontal oscillation was 17° in the visual angle of the mouse; the horizontal spatial cycle of the vertical stripes was 6.4° in the visual angle. First, the temporal frequency was changed from 0.25 Hz to 0.15 Hz or 0.50 Hz while keeping other parameters fixed. Interestingly, the autofluorescence fluctuations followed the changes in the temporal frequency, and a clear sharp amplitude at the frequency locked to the presented visual stimulation was evident in the amplitude spectrum in all conditions (***Figure 5A***). The peak ratio of the autofluorescence was examined for temporal frequency ranging from 0.15 to 0.50 Hz and normalized to that of 0.25 Hz (PR0.25; ***Figure 5D***, left panel). No significant difference was found, showing that vasomotion can lock to any temporal frequency within the range examined. It should be noted that the mice were first trained with 0.25 Hz temporal frequency stimuli, and subsequently tested with multiple temporal frequency stimuli in pseudo-random order and the original 0.25 Hz was tested again sometime during the sequence. The visually induced vasomotion quickly adapted to any new frequency tested and the reversibility of the adaptation was confirmed.

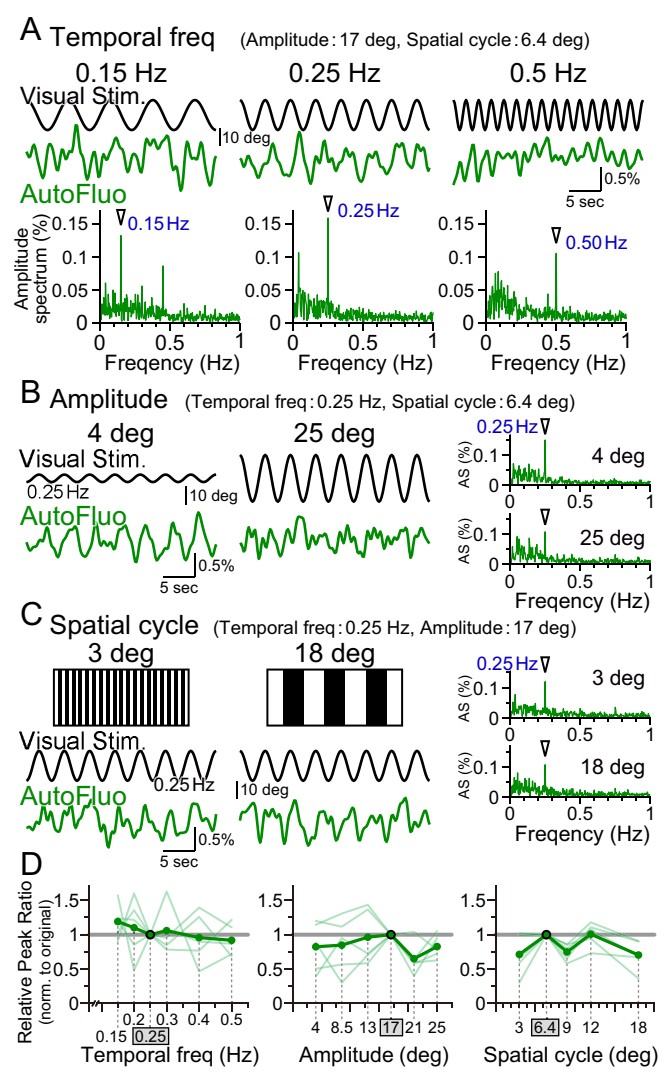

**Figure 5.** Vasomotion entrainment over a wide range of visual stimulation parameters. (**A–C**) Autofluorescence signals at the flocculus in response to visual stimulation with various parameters. In the original condition, the temporal freq (the temporal frequency of horizontal oscillation) was 0.25 Hz, the amplitude (the spatial amplitude of the horizontal oscillation) was 17° in the visual angle of the mouse, and the spatial cycle (the horizontal spatial cycle of the vertical stripes) was 6.4°. All the fluorescence signal data were filtered with 50–1000 mHz. (**A**) The temporal frequency of visual oscillation was varied. The location of the peaks in the autofluorescence signal amplitude spectrogram exactly matched the frequency of the visual stimulus in almost all cases (arrowhead). (**B**) The amplitude of the visual oscillation was varied. (**C**) The spatial cycle of the stripe patterns was varied. (**D**) The relative peak ratio of the autofluorescence signal at the temporal frequency matching that of the presented visual stimulus was calculated. This relative peak ratio was normalized to that of the original condition. Varying the temporal freq, the amplitude, or the spatial cycle almost had no effect on the relative peak ratio. This suggests that vasomotion can tune in to any visual stimulation within the parameter range that we tested. Significant differences were shown only when the spatial cycle was varied (spatial cycle, 6.4° [original] vs. 9°, p=0.0028, *p<0.0125; 6.4° [original] vs. 18°, p=0.015, *p<0.017; one-sample *t*-test with Holm correction for multiple tests). Individual mice's data (as an average from the three sessions of the 5 min test) are shown in pale green, and the average data from the examined mice (n = 5 mice) are shown in dark green.

Similarly, the effect of changing the spatial amplitude of the horizontal oscillations was examined, while keeping other parameters fixed. A large amplitude would require the mouse to move its eyes largely within their eye sockets. The temporal frequency-locked vasomotion was apparent in the low 4° amplitude as well as in the high 25° amplitude (**Figure 5B**). The relative peak ratio normalized to that of 17° amplitude showed that the vasomotion was substantially induced in response to any

amplitude between 4° and 25° (*Figure 5D*, middle panel). Lastly, the horizontal spatial cycle of the vertical stripes was varied (*Figure 5C*). Again, the vasomotion could follow the visual stimuli irrespective of the fineness of the presented stripes (*Figure 5D*, right panel). The combination of parameters of the visual stimuli used in the rest of the article was well within the adequate range that would induce maximal visually induced vasomotion.

## Vasomotion is not the consequence of eye movements

We have shown above that the bulk dynamics of vasomotion under the optical fiber could be quantified with autofluorescence intensity fluctuations using fiber photometry. HOKR visual stimulation induces both eye movement and frequency-locked vasomotion. Here, we examined whether the eye movement itself is the cause of vasomotion.

First, the right eye trace in response to the horizontally oscillating visual stimulus was examined (*Figure 6A*). The eye tracked the visual stimulation well and there was virtually no phase shift in the eye trace relative to the visual stimulation cycle (*Figure 6B*). The TexasRed signal fluctuation in the flocculus in the right hemisphere was oscillatory with the visual stimulation; however, the phase of the TexasRed signal was either 0° or 180° relative to the visual stimulation, depending on the animal (*Figure 6C*). This indicates that in the 0° phase-shifted animals, the maximum vasodilation and vasoconstriction occur when the right eye is at the maximal nasal and temporal locations, respectively. Conversely, in the 180° phase-shift animals, it is the other way around. In either mouse, the maximum vasodilation and vasoconstriction occur when the speed of the eye movement reaches zero. The reason why the phase of the vasomotion becomes fixed to 0° or 180°, depending on the animal, is unresolved.

As the vasomotion apparently followed the visual stimuli unbelievably well, we suspected several artifacts affecting the fluorescence recording. First, we suspected that the visual stimulation itself is penetrating the back of the eye, through the brain, and to the flocculus of the cerebellum. Thus, the photo intensity detected with the optical fiber could be coming directly from the oscillatory visual stimulus. To address this possibility, the mouse was anesthetized by isoflurane (*Figure 6D*). Under anesthesia, the eye fails to follow the screen. The vasomotion was detected with the 'shadow' imaging method by studying the dYFP fluctuations. No oscillatory signals were detected in the dYFP. Next, in the TexasRed injected mouse, the HOKR visual stimulation was presented, which induced a nice eye movement following the visual stimulus (*Figure 6E*). However, in this case, we did not send the excitation light through the optical fiber but kept the PMT active. If the visual stimulus on the presented screen is somehow reaching the optical fiber, then it should be detected in this condition, but it was not.

Finally, we investigated the possibility that the fluorescence signal fluctuation is due to the motion artifact. The physical eye movement could produce a movement in the brain tissue and the optical fiber location relative to the blood vessels could be moved, leading to the fluctuation of the fluorescence signals. When the mouse is presented with visual stimulation, in the trained mouse, the eye usually follows the visual stimulus completely. However, even during a single session, the frequency-locked TexasRed signal was not always apparent. In an example shown in *Figure 6F*, the eye trace was completely in sync with the visual stimulation either in the left panel segment or in the right. However, TexasRed signal fluctuation with an amplitude spectrogram of 0.25 Hz was not apparent in the left panel segment but was significant in the right panel segment. This observation argues against the possibility that the detected fluorescence fluctuation is the direct result of eye movement. The amplitude of the oscillatory eye movement was plotted against the PR0.25 of the TexasRed during a single visual session (*Figure 6G*). The cycle of visual stimulation producing either a large or small eye movement did not always result in a large or small amplitude fluctuation of the vasomotion, respectively, and no correlation was found.

## Possibility of the contribution of vasomotion entrainment to the HOKR performance

Repeated HOKR training usually results in a larger amplitude of the eye oscillations, which reaches close to the actual visual presentation amplitude. We have shown above that the frequency-locked vasomotion in the cortex also got entrained with repeated training sessions (*Figure 3*). It has been shown that HOKR performance increase relies on the neuronal and glial activity in the cerebellar

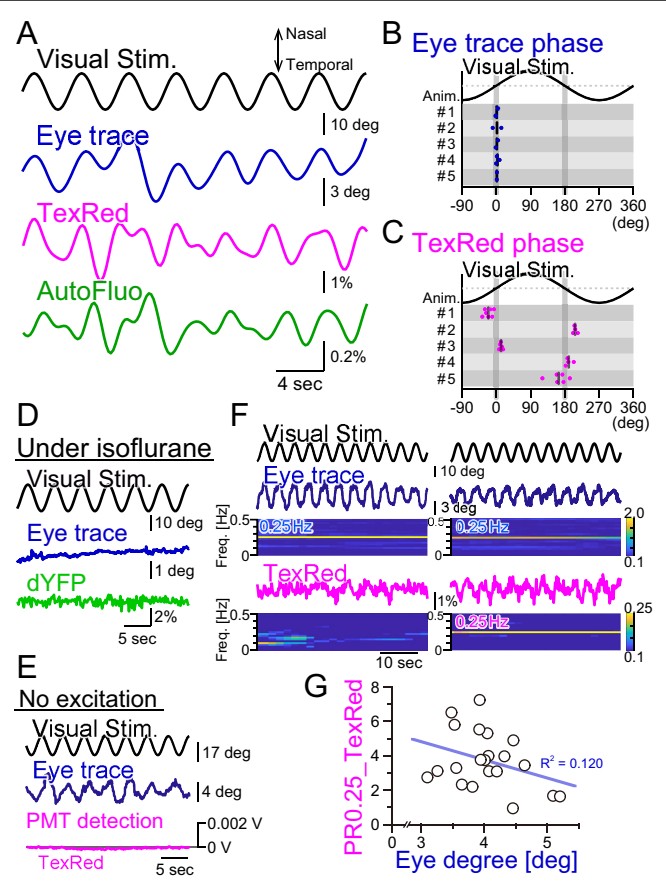

**Figure 6.** Fluorescence fluctuations are not the results of eye movement artifact. (**A**) An example of the visual stimulation time course, the eye trace, the TexasRed signal, and the autofluorescence signal fluctuations are shown. The data was filtered with 50–500 mHz. (**B**) The phase of the eye trace relative to the visual stimulus. The visual stimulus was presented with a 4 s cycle (0.25 Hz), thus, the eye trace data was segmented with 4 s durations during the first 5 min of each training. The segmented data were aligned, averaged, and fitted with a sine curve. The phase difference between the visual stimulus and the fit to the averaged eye trace was calculated. The mean of the phase in each animal across multiple sessions is shown as a vertical black bar. The average phase lag was 2.3 ± 1.0° in five animals, indicating that the eye trace synchronize nearly completely with the visual stimulus. (**C**) The phase of the TexasRed signal fluctuations relative to the visual stimulus. The average TexasRed signal fluctuation with a 4 s cycle was calculated from segments extracted from several sessions, each with 1–6 min in duration. The phase lag of the TexasRed signal relative to the visual stimulus was clustered either at 0° or 180°. This shows that the local brain blood volume in the right flocculus becomes the largest when the right eye is either in the most nasal position or the temporal position. (**D**) Under anesthesia with isoflurane, the eye movement was not induced. The dYFP signal from the YCnano50 expressed in astrocytes also did not oscillate. (**E**) With no excitation light sent to the optical fiber, the PMT detected no signal (0 V). The presented visual stimulus did not reach the fiber optics to directly induce oscillatory signals. (**F**) The amplitude of the eye movement was normally nearly constant during a single session; however, slightly strong (left) and weak (right) eye movements were observed. The TexasRed signal fluctuation at the frequency locked to the visual stimulus was not constant during the session and PR0.25 of the TexasRed signal could be seen in some (right) but not the other (left) segments of the session. (**G**) PR0.25 of TexasRed signal was plotted against the oscillatory eye movement amplitude in 40 s segments within a single visual stimulation session (15 min). The first segment was excluded (n = 21 segments total). No correlation was found between these two factors.

flocculus (**Kanaya et al., 2023**). We next evaluated whether the plasticity of the vasomotion dynamics in the flocculus correlates with the HOKR performance increase.

HOKR-spaced training paradigm was used. Briefly, 15 min training sessions were repeated at 1 hr intervals four times on day 1 (i.e., the training day; **Figure 7A**). Averaged amplitudes of the eye movement during the first and last 3 min of each training session were calculated. The calculated averaged

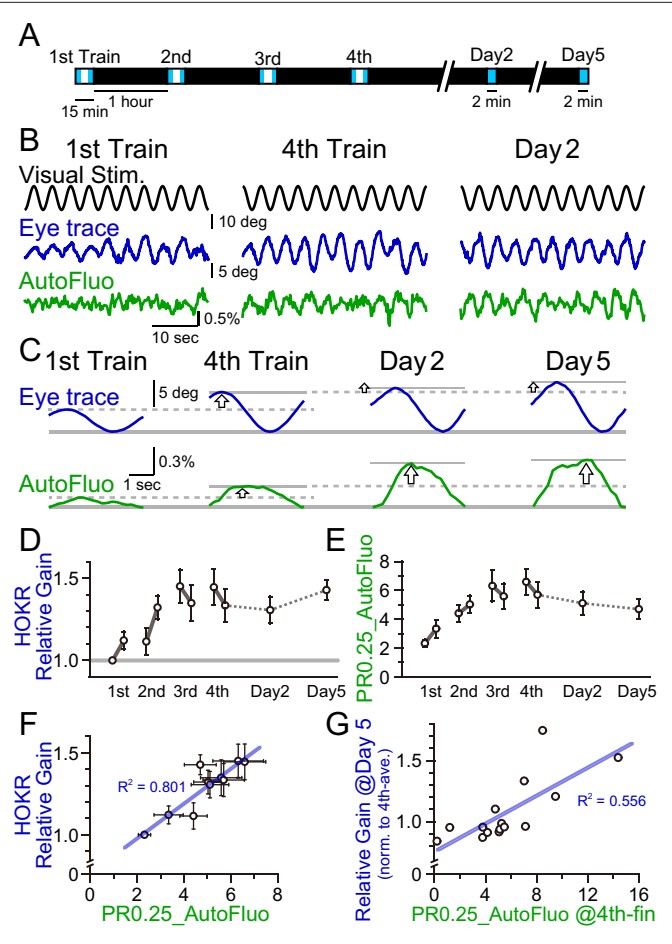

**Figure 7.** A concomitant increase in the horizontal optokinetic response (HOKR) performance and vasomotion entrainment. (**A**) Schematic schedule of the HOKR spaced training paradigm. The 15 min training sessions were repeated at 1 hr intervals for four times on day 1 (the training day). The average amplitudes of the eye movement during the first and last 3 min of each training session were analyzed (during the period indicated by the blue box in the schematics). The 2 min tests measuring long-term memory were done on days 2 and 5. (**B**) Representative traces of the visual stimulus, the eye trace, and the autofluorescence during the first and the fourth training on day 1 and the test done on day 2. The amplitude of the eye movement and the vasomotion gradually increased in amplitude with repeated sessions. (**C**) The eye traces and the autofluorescence traces were segmented in 4 s durations, corresponding to the visual stimulation frequency of 0.25 Hz, and aligned and averaged. (**D**) The amplitude of the eye movement relative to that of the start of the first training session on day 1 was refined as the HOKR relative gain. Average HOKR relative gain across n = 15 animals is plotted (mean ± SEM). (**E**) PR0.25 of the autofluorescence signals was averaged across n = 15 animals and plotted against sessions. (**F**) Averaged HOKR relative gain was plotted against the averaged PR0.25 of autofluorescence signals. Error bars indicated S.D. (n = 15 animals). (**G**) HOKR relative gain on day 5 normalized to that on the fourth training on day 1 was plotted against the PR0.25 of autofluorescence signals at the end of the fourth training on day 1 (n = 15 animals).

eye amplitudes were normalized to the initial amplitude measured at the first 3 min of the first training session. This value was termed the 'relative gain'. A brief 2 min test was done on day 2 and 5 to evaluate the long-term changes in the performance. Along with the HOKR eye movement measurements, autofluorescence signals were measured with the optical fiber implanted close to the flocculus. The amplitude of the autofluorescence signal at a temporal frequency locked to the visual stimulation frequency (0.25 Hz) was analyzed. As above, this amplitude was divided by the mean amplitude in the 0.1–1 Hz range to derive the PR0.25. As the autofluorescence fluctuations were often noisy, the autofluorescence signals during the first and last 5 min of each training session were used for the PR0.25 calculation. The long-term memory tests done on days 2 and 5 were only 2 min in duration, thus, the full 2 min were used for the PR0.25 calculations.

As described previously, HOKR performance increased with repeated training sessions during day 1 (*Figure 7B-D*). With the current experimental conditions, the average HOKR performance tested on days 2 and 5 was nearly the same as the performance on the last session of day 1; however, long-term changes in the performance were sometimes apparent in individual cases (*Figure 7C*). The PR0.25 of the autofluorescence signal changes with the spaced training protocol basically followed the plastic changes of the eye movement (*Figure 7E*). PR0.25 tended to increase with repeated training sessions.

The HOKR relative gain was calculated for each test (i.e., 10 tests total; the early and the late time tests of each of the four sessions on day 1, and tests on days 2 and 5), and the HOKR relative gain was averaged across all animals examined (n = 15 mice). Similarly, the PR0.25 in autofluorescence fluctuations were averaged across animals in 10 tests. The average HOKR relative gains were plotted against the average PR0.25 values and a strong correlation was found (*Figure 7F*; each data point represents an average score of n = 15 mice and 10 data points are shown for the 10 tests). In addition, it should be pointed out that a large variance was found both in the HOKR relative gain and the PR0.25 of the autofluorescence fluctuations across animals (X and Y error bars indicate S.D. in *Figure 7F*). Therefore, in individual animal recordings, the strong correlation between the HOKR relative gain and the PR0.25 of the autofluorescence fluctuations may not always be apparent. It is possible that the plasticity governing HOKR and vasomotion increases could be induced by a parallel process.

Correlations between HOKR and vasomotion across individual animals were searched. HOKR relative gain on day 5 normalized to that for the fourth training on day 1 was plotted against the PR0.25 of the autofluorescence fluctuations during the last 5 min of the fourth training session on day 1 (*Figure 7G*). Interestingly, a moderate but significant correlation between these parameters was found (n = 15 mice). This may suggest that the level of visually induced vasomotion contributes to the consolidation of HOKR performance in later days. These results imply a complex and intimate relationship between vasomotion and neuronal circuit plasticity, learning, and memory.

## Discussion

We show that visually induced vasomotion can be frequency-locked to the visual stimulus and can be entrained with repeated trials. The initial drive for the vasomotion, or the sensory-evoked hyperemia, must be coming from the neuronal activity in the visual system. The visually induced vasomotion is likely triggered by activation of the neurovascular interaction (*Kayser et al., 2004*; *van Veluw et al., 2020*). Surprisingly, the entrained vasomotion was observed not only in the visual cortex but also widely throughout the surface of the brain and deep in the cerebellar flocculus. The global entrainment could be realized through separate mechanisms from the local neurovascular coupling. What is also unknown is where the plasticity occurs. The neuronal visual response in the primary visual cortex could potentially decrease with repeated visual stimulation presentation as the adaptive movement of the eye should decrease the retinal slip. With repeated training sessions, a more static projection of the presented image will likely be shown to the retina. Repeated visual stimulus may induce synaptic plasticity not in the visual system but at the neuronal projections from the retina to a central nucleus that controls a brain-wide vasomotion, such as in the autonomic nerve system (*Korte et al., 2023*). Instead, the neurovascular coupling could be enhanced with increased responsiveness of the vascules and vascular-to-vascular coupling could also be potentiated.

We show that the fluorescence signal from the brain parenchyma follows the vasomotion because the blood vessels largely lack fluorescence signals within the filter band that we observe. This is described as 'shadow imaging'. What was rather puzzling was that autofluorescence signals were phase-shifted in time. This suggests that these autofluorescence signals have an anti-phase 'shadow imaging' component and another component that is phase-shifted in time. Glucose and oxygen are likely to be abundantly delivered during the vasodilation phase compared to the vasoconstriction phase of vasomotion. These molecules will trigger cell metabolism and endogenous fluorescent molecules such as NADH, NADPH, and FAD may increase or decrease with a certain delay. These molecules are required for biochemical reactions to occur. Therefore, the concentration fluctuation of these metabolites could lag in time to the changes in the blood flow. It is also expected that these metabolites may fluctuate according to the neuronal activity that triggers visually induced vasomotion or functional hyperemia.

The stress in head restraint mice could well affect spontaneous vasomotion as well as visually induced vasomotion. Whether the experimentally induced increase in stress would interfere with the

vasomotion or not could also be studied. With the TexasRed experiments, we observed that tail-vein injection stress appeared to interfere with the HOKR learning process. In the experiments presented in *Figure 3*, TexasRed was injected before session 1. Vasomotion entrainment likely progressed with session 2 and 3 training. Before session 4, TexasRed was injected again to visualize the vasomotion. The oscillatory vasomotion was clearly observed in session 4, indicating that the stress induced by tail-vein injection could not interfere with the generation of visually induced vasomotion.

It is not possible to distinguish whether the visually induced oscillatory vasomotion observed in this study was driven by the mechanisms of spontaneous vasomotion or functional hyperemia. Both of them may be involved in the visually induced vasomotion. The spontaneous vasomotion requires the coordinated activity of the vascular SMCs. Oscillatory release of calcium ions from the sarcoplasmic reticulum results in oscillations in the membrane potential of SMCs. Via the gap junctions between SMCs, synchronized oscillations are created (*Haddock et al., 2006*; *Haddock and Hill, 2002*; *Hill et al., 1999*; *Peng et al., 2001*; *Schuster et al., 2001*). Blood flow in the central nervous system was assumed to be regulated primarily by the SMCs in precapillary arterioles; however, capillaries lacking smooth muscle have been shown to be controlled by pericytes (*Peppiatt et al., 2006*). The spontaneous vasomotion observed in our study could be the result of constriction and dilation of arterioles or capillaries controlled by SMCs or pericytes, respectively. Astrocytes also constitute the neurovascular unit; however, whether astrocytic calcium is involved in controlling local cerebral blood flow is in question (*Ozawa et al., 2023*). It is still possible that astrocytes could regulate vasomotor amplitudes via stretch-mediated feedback (*Bazargani and Attwell, 2016*; *Haidey et al., 2021*). Spontaneous vasomotion has been suggested to be driven by the periodic neural activity in the resting state (*Martuzzi et al., 2009*; *Mateo et al., 2017*). Neuronal projections from the neuronal nuclei projecting to the whole brain may be controlling and synchronizing vasomotion via neurovascular coupling (*Korte et al., 2023*). It is also possible that ultra-slow vasomotion could be driven not only by the neuronal rhythms but also by the oscillatory systemic changes, such as blood pressure and carbon dioxide in the blood (*Panerai et al., 1998*; *Sassaroli et al., 2012*; *Wise et al., 2004*).

Functional hyperemia has been shown to be induced by visual stimulation (*Heeger et al., 2000*; *Kayser et al., 2004*; *Niessing et al., 2005*; *Rivadulla et al., 2011*; *Shaw et al., 2021*; *Stickland et al., 2019*). Visual stimulation evokes neuronal action potential firing and through the neurovascular coupling mechanisms, vasomotion is affected. If a pulsatile visual stimulus is periodically presented, that would result in vasomotion fluctuation with the same frequency as the presented visual stimulus (*Kayser et al., 2004*; *van Veluw et al., 2020*). Previously, such visual stimulus-evoked vasomotion was reported to be restricted to the visual cortex (*van Veluw et al., 2020*). However, in our study, we show that vasomotion entrainment occurs with repeated HOKR visual stimulation, and the coordinated vasomotion, which is frequency-locked to the visual stimulus, was observed not only in the visual cortex but also widely throughout the surface of the cortex and the cerebellum, and also deep in the flocculus region of the cerebellum. It is possible that vasomotion entrainment is induced in other areas as well, including the limbic system and the brain stem.

Vasomotion could be beneficial for regulating the blood flow and the vascular surface area, which would affect the supply of nutrients (e.g., glucose) and oxygen in the brain. Mathematical modeling has shown that effective vascular resistance decreases with vasomotion (*Meyer et al., 2002*). Low-frequency vasomotion has been shown to be beneficial for tissue oxygenation, especially in regions where perfusion is restricted (*Davis et al., 2008*; *Nilsson and Aalkjaer, 2003*). As astrocyte processes surround the blood vessels, the coordinated activity of astrocytes is presumed to be important for the efficient delivery of the energy source to neurons for information coding and plasticity. Furthermore, it was recently reported that vasomotion could also drive paravascular clearance or intramural periarterial drainage (IPAD) (*van Veluw et al., 2020*). Initially, arterial pulsation from the heart was considered to be the driving force for the IPAD; however, simulation modeling suggested that vasomotion was required for the efficient IPAD (*Aldea et al., 2019*).

40 Hz sensory stimulation has been proposed to be potentially beneficial for the treatment of Alzheimer's disease (AD) (*Adaikkan et al., 2019*; *Iaccarino et al., 2016*; *Martorell et al., 2019*). However, it was recently claimed that the 40 Hz flickering light does not entrain gamma oscillations and suppress amyloid beta (Aβ) accumulation in the brains of the AD model mouse (*Soula et al., 2023*). In another study, it was shown that flashing visual stimuli could increase the amplitude of vasomotion, which resulted in an increase in the paravascular clearance rates in the visual cortex of awake

mice (*van Veluw et al., 2020*). In mice with cerebral amyloid angiopathy, impaired paravascular clearance and Aβ accumulation occur (*Charidimou et al., 2017*; *Peng et al., 2016*). It was also suggested that low-frequency arteriolar oscillations could drive the drainage of soluble waste products (*van Veluw et al., 2020*).

In our study, we show that the HOKR-driving oscillatory visual stimulus, which is much slower (0.25 Hz) than the 40 Hz, is effective in producing a brain-wide oscillatory vasomotion. In addition, repeated presentation with the spaced training protocol was effective in driving the plastic vasomotion entrainment. Paravascular clearance and drainage of waste products, such as Aβ, may be enhanced with vasomotion entrainment. We show that synaptic plasticity change related to cerebellar motor learning is enhanced in parallel with the vasomotion entrainment. It is possible that vasomotion enhancement may increase the meta-plasticity and provide a basis for efficient neuronal plasticity to occur. In the cerebellum, neuron-to-astrocyte (or Bergmann glia) (*Matsui et al., 2005*; *Matsui and Jahr, 2004*; *Matsui and Jahr, 2003*) and astrocyte-to-neuron (*Beppu et al., 2021*; *Beppu et al., 2014*; *Sasaki et al., 2012*) signaling pathways have been identified and such neuron-astrocyte interaction has been shown to be essential for the online HOKR learning to occur (*Kanaya et al., 2023*). Late offline HOKR learning was dependent in part on the phagocytosis activity of astrocytes (*Morizawa et al., 2022*). It is possible that the energy required for the long-term plastic changes in the brain is supported by the cooperative actions of the astrocytes and the entrained vasomotion. Aβ has been shown to constrict capillaries and the resulting slight hypoxia and neuronal damage may actually be the cause of dementia in AD (*Nortley et al., 2019*). The slow sensory stimulation may possibly be effective for AD treatment; however, evidence of enhanced Aβ drainage and cognitive decline resistance with such sensory stimulus using AD model mice would be required in the future to support such a claim.

## Materials and methods

### Animals

All experiments were performed in accordance with the recommendations of the Regulations for Animal Experiments and Related Activities at Tohoku University (2019LsA-017-10). Efforts were made to minimize animal suffering and reduce the number of animals used. Mice were housed individually for more than 7 d under standard laboratory conditions with food and water ad libitum.

C57BL/6NJcl male mice (2–4 months old) were used in this study in order to reduce the variance of HOKR performance among animals (*Aziz et al., 2014*). $Mlc1$-$tTA::tetO$-$YC_{nano50}$ double transgenic mice were also used, which express the tetracycline transactivator (tTA) under the astrocyte-specific *Mlc1* promoter (*Tanaka et al., 2012*). The expressed tTA activates the tetO promoter and the FRET-based $Ca^{2+}$ sensor $YC_{nano50}$ is expressed specifically in astrocytes, including the cerebellar Bergmann glial cells (*Kanemaru et al., 2014*). Astrocyte $Ca^{2+}$ dynamics associated with vasomotion were of potential interest to study; however, HOKR learning could not be reliably induced in $Mlc1$-$tTA::tetO$-$YC_{nano50}$ transgenic mice for unknown reasons. Thus, we concentrated on the vasomotion dynamics resolved by the autofluorescence detection for most of the article. Only the YFP part of the CFP-YFP conjugated protein of $YC_{nano50}$ was used to assess the 'shadow' imaging principle for detecting vasomotion dynamics.

### Wide-field macro-zoom microscope

For surgical preparation, mice were anesthetized under ~5% isofluorane for induction and 0.5–2.0% isoflurane mixed with the air for maintenance. The scalp was cut open and the periosteum was gently removed to expose the crania. A stainless steel chamber frame (CF-10; Narishige, Tokyo, Japan) was used as a holding jig to place the head in place under the microscope. The chamber frame was attached to the edges of the exposed crania of the mouse to avoid obstruction of the view of the cortex and the cerebellum using a synthetic resin (Super bond C&B; Sun Medical Company, Shiga, Japan) mixed with charcoal particles. In the thinned skull imaging (*Drew et al., 2010*), the skill was gently thinned with a drill with precautions not to penetrate the skull. In both thinned skull and intact skull imaging (*Morii et al., 1986*), the crania were covered with a UV-curing resin (Jelly Nail; IML Inc, Tokyo, Japan) to prevent desiccation of the bone. The resin greatly improved the transparency of the skull, and it could protect the field of view for over weeks.

The resin and the short UV exposure are commonly applied to human fingernails without health concerns. Even after the short UV exposure for curing resin, fluorescence proteins in the mouse brain were not photobleached. A tube made of black aluminum was attached to the top of the mouse's head so as not to obstruct the view of the microscope. All gaps between the head and this custom-made tube were sealed with the Jelly Nail resin with charcoal particles to prevent light from the video monitor used for HOKR visual stimulus from leaking in. The mice were allowed to recover from the surgery for at least 3 d.

Prior to setting the mouse under the microscope, the mouse was intravenously injected via the tail vein with TexasRed dextran (70,000 MW, Invitrogen; prepared 25 mg/kg in saline at 0.1 ml). Immediately after the injection, the mouse was mounted on a chamber holder (MAG-1 and MAG-A; Narishige) using the chamber frame fixed to its skull. The mouse's body was loosely restrained with a plastic cylinder. The head-fixed mouse was set under a wide-field macro-zoom microscope (MVX10; Olympus, Tokyo, Japan) equipped with an image splitting optics (W-VIEW GEMINI A12801-01; Hamamatsu) and a digital CMOS camera (ORCA-Flash4.0 V3; Hamamatsu Photonics, Shizuoka, Japan). An inverted cone made of a black opaque sheet was attached to the objective lens to prevent the outside light from leaking in and prevent the fluorescence excitation light from the microscope from reaching the mouse's vision and startling the mouse. The inverted cone was capped by the tube attached to the mice's head, thus, the light insulation was complete.

The excitation light to the microscope was provided by an LED-based light source (Niji; Blue Box Optics, Blackwood, UK). For excitation of the TexasRed, excitation of the autofluorescence, and direct excitation of the YFP (dYFP) of the $YC_{nano50}$, were accomplished with 560 nm LED light source with a bandpass filter of ET580/25x (Chroma, VT), 440 nm LED light source with a bandpass filter ET430/24x (Chroma), and 490 nm LED light source with a bandpass filter ET505/20x (Chroma), respectively. A multi-band beamsplitter (69008bs; Chroma) was used to send multiple excitation wavelength lights to the mouse while allowing the emission lights of the TexasRed, the autofluorescence, and the dYFP to pass through. When necessary, the emission light of the TexasRed and that of the autofluorescence or the dYFP were split using a dichroic mirror (T590lpxr, Chroma). The wavelength of the emission lights was restricted using a custom bandpass filter (passband: 473 ± 12 nm and 625 ± 15 nm, Chroma), a bandpass filter ET540/40m (Chroma), or a bandpass filter ET540/30 m (Chroma) for the detection of TexasRed, autofluorescence, or dYFP, respectively.

The frame exposure time of the camera was 89 ms and frames were recorded at 5 Hz sampling frequency. The magnification of the microscope was ×0.8 or ×6.3. All settings of the camera were controlled by the HCImageLive software (Hamamatsu). The power of excitation light at the wavelengths of 560 nm and 440 nm delivered from Niji were 1.7 mW and 3.2 mW with ×0.8 magnification, respectively. The power of excitation light with the wavelength of 560 nm, 440 nm, and 490 nm with ×6.3 magnification were 8 mW, 13 mW, and 4 mW, respectively. In all of the recording configurations, each excitation light was sequentially delivered in 91 ms pulses with 109 ms intervals.

## Fiber photometry

Prior to surgical implantation of the optical fiber, the mouse was anesthetized with three types of mixed anesthesia, consisting of 0.75 mg/kg of medetomidine hydrochloride (Domitol; Nippon Zenyaku Kogyo Co., Ltd, Fukushima, Japan), 4 mg/kg of midazolam (Midazolam, Sandoz Inc, Japan), and 5 mg/kg of butorphanol tartrate (Vetorphale; Meiji Seika Pharma Co., Ltd, Tokyo, Japan). A screw (6 mm head diameter and 10 mm length) was attached to the cranial bone with its head down between bregma and lambda using a synthetic resin (Super bond C&B; Sun Medical Company). This screw was used as a holding jig to keep the mouse's head fixed toward the visual stimulus presenting monitor. A glass optical fiber (core diameter 400 μm, 0.39 NA; FT400UMT; Thorlabs, NJ) was implanted in the cerebellum close to the flocculus (coordinates X = 2.0 mm, Y = –5.65 mm, Z = 2.9 mm, $\alpha$ = 10°). This optical fiber was fixed using a synthetic resin (Super bond C&B) mixed with charcoal particles to prevent external lights from penetrating the resin. For recovery from three types of mixed anesthesia, 0.75 mg/kg of atipamezole hydrochloride (Antisedan; Zenyaku Kogyo Co., Ltd, Tokyo, Japan) was administered. The mice were allowed to recover from surgery for at least 5 d. When necessary, the mice were intravenously injected via the tail vein with TexasRed dextran prior to recording. Each mouse was head-restrained by tightening the screw on the head of the mouse to the custom-made apparatus and its body was loosely restrained in an upward-sloping plastic cylinder.

A custom-made optical apparatus for fiber photometry was used (Lucir, Tsukuba, Japan). Excitation lights were delivered from the SPECTRA X Light Engine (Lumencor, OR). An optical fiber from the fiber photometry apparatus was connected to the optical fiber implanted in the mouse with a ferrule-to-ferule connection using a split sleeve. Excitation light was sent to this optical fiber and the power of excitation light with the wavelength of 575 nm, 510 nm, and 440 nm were 3 mW, 11 µW, and 2.2 mW, respectively, at the tip of the implanted optical fiber. The emitted light came back through the optical fiber, reached the fiber photometry apparatus, and was detected using the two photomultiplier tubes (PMTs) (PMTH-S1-CR316-02; Zolix Instruments Co., Ltd, China) with current-to-voltage amplifiers (HVC1800; Zolix). The electronic signals from the PMTs were digitized with Micro1401-3 plus ADC12 top box (CED, Cambridge, UK) and recorded using the Spike2 software (CED) at a 5 kHz sampling frequency. In all of the recording configurations, each excitation light was delivered in 20 ms pulses with 180 ms intervals. The signals were averaged within the 20 ms pulse, resulting in an effective sampling frequency of 5 Hz.

TexasRed and dYFP were excited with the wavelength of 575 nm and 510 nm, respectively, and the excitation wavelengths were restricted with a multi-bandpass filter 69008x (Chroma). Specifically, the light for directly exciting YFP was delivered via the relevant band of 503 nm/19.5 nm (FWHM) of the multi-bandpass filter 69008x. With this filter, negligible light with a wavelength below 490 nm is passed through. The excitation wavelength of CFP is below 490 nm, and thus excitation of CFP can be assumed to be negligible with the light used for directly exciting YFP. A multi-band beamsplitter (ZT445/514/594rpc; Chroma) was used to send multiple excitation lights to the mice and pass through the emission lights of TexasRed or dYFP. The emission lights of TexasRed and dYFP were split by a dichroic mirror, FF605-Di02 (Semrock, IL). The wavelength of the emission lights was restricted with a bandpass filter ET640/30x (Chroma) for TexasRed and with ET539/21x (Chroma) for dYFP detection.

For excitation of TexasRed and autofluorescence, lights with the wavelength of 575 nm and 440 nm restricted by a multi-bandpass filter 69008x (Chroma) were used. A multi-band beamsplitter 69008bs (Chroma) was used to send multiple excitation lights to the mice and pass through the emission lights of TexasRed and autofluorescence. The emission lights of TexasRed and autofluorescence were split by a dichroic mirror FF605-Di02 (Semrock). The wavelength of the emission lights was restricted with a bandpass filter ET645/75m (Chroma) for TexasRed and with ET537/29m (Chroma) for autofluorescence detection.

For excitation of only the autofluorescence, light with the wavelength of 440 nm restricted with a bandpass filter ET436/20x (Chroma) was used. A dichroic mirror T495lpxr (Chroma) was used to send the excitation light to the mice and pass through the emission light of the autofluorescence. The wavelength of the emission light was directly restricted with a bandpass filter FF01-550/88 (Semrock) for autofluorescence detection.

## Visual stimulus presentation

For visual stimulation, sinusoidal oscillations of vertical stripes displayed on two PC monitors arranged orthogonally were presented to the animal. The visual stimuli were created and controlled with the ImageJ/Fiji software (National Institute of Health, MD) or the custom program written in MATLAB (MathWorks, Natick, MA), for use with the macro-zoom microscope or the fiber photometry, respectively. In experiments other than the ones shown in *Figure 5*, the amplitude of the oscillations was 17° in the visual angle of the mouse, the temporal frequency of the oscillations was 0.25 Hz, and the spatial cycle of the displayed stripe pattern was 6.4°.

## Analysis of the eye movement for the HOKR experiments

In fiber photometry experiments, the eye movements were simultaneously recorded. The method of recording and analyzing eye movement was previously described in detail (*Kanaya et al., 2023*). During the HOKR experiments, the mouse was exposed to white noise sounds, which helped keep the mouse alert. An angled hot mirror, which reflects only the IR light, was placed between the mouse's right eye and the monitor, thus the visual field of the mouse was unobstructed. The monitored eye (the right eye) was illuminated with two infrared (IR) LEDs. The corneal reflections of the IR LEDs were used as the reference point. The reflected IR image of the eye was captured using a CCD camera (DN3V-30BU; Shodensha Inc, Osaka, Japan) with the IR-cut filter removed.

The captured video taken at 30 fps was analyzed using DeepLabCut (*Mathis et al., 2018*), and the center of the pupil was tracked (see *Kanaya et al., 2023* for details). The angle of the eye direction was calculated based on the pupil and camera position using a previously introduced method (*Sakatani and Isa, 2004*). When the difference in the center eye position between two consecutive frames (1/30 s) was more than 1° in visual angle, it was assumed that a saccadic eye movement had occurred. These incidents of saccadic movements were removed. Drifts in the eye movement were eliminated by a bandpass filter with a 100–500 mHz frequency. The amplitudes of the eye were calculated by the difference between the averaged maximum peak and minimum peak. HOKR relative gain was calculated by normalizing the amplitude of the eye movement at the start of the first training session.

## Vessel diameter and vasomotion analysis with macro-zoom microscope images

Images acquired with the macro-zoom microscope were analyzed with ImageJ/Fiji and AxoGraph (Axograph Scientific, New South Wales, Australia) software. The blood vessels focused and studied were likely to be arteries or arterioles found on the surface of the brain which were filled with TexasRed fluorophore via intravenous injection. The blood vessel in focus was rotated so that the flow of the blood would be aligned with the vertical axis. The stack of images taken as time series was cropped with the blood vessel in the center of the image, and this stack was named the 'original stack'. The background intensity was measured within the ROI on the left and right of the vessel and an average value was calculated for each image of the stack. A new stack with the same XY pixel dimensions as the '(1) original stack' was created and each image of the stack had a single uniform value of the background intensity calculated above. This background stack was subtracted from the '(1) original stack' to create the '(2) background subtracted stack'. Due to the dilution of the TexasRed fluorophore and the photobleaching effect, the fluorescence in the blood vessels tends to fade in time. Thus, a normalization procedure to counter the fading effect was developed. First, using the above '(2) background subtracted stack', an ROI was set well within the blood vessels and a time-series data of the average intensity within the ROI was calculated. This data was strongly low-pass Gaussian filtered at 5 mHz, and the transient fluctuations of intensity due to vasomotion were totally eliminated, leaving only the baseline decline of intensity due to the fading effect was left. Again, a new stack with the same XY pixel dimensions as the '(1) original stack' was created and each image of the stack had a single uniform value of the baseline intensity calculated above. The '(2) background subtracted stack' was divided by this baseline stack to create the '(3) baseline normalized stack'. With this procedure, the images are normalized to the basal TexasRed intensity in the blood vessels. TexasRed intensity differs among vessels and trials, thus this procedure is effective in the normalization of multiple experiments. The '(3) baseline normalized stack' was resliced with a line vertical to the vessel using the ImageJ/Fiji software and the temporal change of the TexasRed profile along this line was created. Multiple lines vertical to the vessel were averaged, allowing a better signal-to-noise ratio. This data was low-pass Gaussian filtered (~0.05 pixel) in each slice. The left side and the right side of the vessel often had different background intensity levels. Therefore, these cross-sectional profiles of the blood vessels were processed with a flat-sloping baseline function in AxoGraph to eliminate this horizontal slope.

The cross-sectional profile of an undilated vessel in the basal phase was referred and 10% of the maximum value in the profile was calculated. The full width of the cross-section profile at this threshold value was defined as the vessel diameter. Since a confocal or two-photon microscope was not used, an ideal optical section image could not be obtained. Therefore, the fluorescence intensity of the line profile across the vessel would be expected to increase toward the center of the vessel as the thickness of the vessel increases. One would not expect to see a square line profile. Therefore, full width at half-maximum value of the line profile would likely be an underestimate of the actual vessel diameter. Approximately 10% is the minimum intensity that we could distinguish from the background intensity fluctuations. Full width at 10% maximum should be considered just an index of the actual diameter, and the 'true' diameter of the vessels could not be determined. In most of the current study, we dealt only with the change of the vessel diameter relative to the basal diameter. The calculated vessel diameter at each time point was normalized to the vessel diameter in the undilated phase.

The vasodilation events were detected using the event detection function in AxoGraph. Thresholds were determined arbitrarily as the amplitudes of the vasodilation varied across the vessels. Inter-event interval and peak amplitudes were calculated. Unit dilation time was defined as the full-width time at

the half-maximum of the amplitude of the vasodilation. Inter-event interval, peak amplitude, and unit dilation time were transformed into probability density function and the cumulative relative frequency was calculated.

### Frequency analysis

Essentially, the same procedure was used for analyzing the temporal frequency profile of the fluorescence signal data recorded from either the macro-zoom microscopy or the fiber photometry. First, the fluorescence fade was corrected and the baseline value of the fluorescence signals was normalized. Using MATLAB, the original fluorescence signal data taken with a sampling frequency of 5 Hz were divided by their own low-pass filtered data (the passband was 50 mHz and the stopband was 10 mHz). The result was subtracted by 1 and multiplied by 100 to transform the data into a percentage display. 5 min or 15 min of data taken during visual stimulus presentation were processed by fast Fourier transform (FFT). The amplitude spectrum and spectrogram were created by executing FFT or short-term Fourier transform (STFT), respectively, by custom programs written in MATLAB. For the spectrogram; the window size was 120 s, 60 s, or 40 s, and the step size was 10 s, 4 s, or 2 s.

The magnitude of the fluorescence signal frequency locked to the presented temporal frequency of the visual stimulus was expressed as the peak ratio of 0.25 Hz (PR0.25). For deriving the PR0.25, first, the peak amplitude of the fluorescence signal at 0.25 Hz was calculated. Next, the mean amplitude of the signal in the 0.1–1 Hz range was calculated. PR0.25 is the signal amplitude at 0.25 Hz divided by the mean at 0.1–1 Hz.

The whole brain surface map of the PR0.25 magnitude presented in *Figure 3F* was created using the following methods. The raw stacked image data taken during the visual stimulus presentation were first processed by using a Gaussian filter with $\sigma = 165$ µm in the x and y dimensions. The time-series data of each pixel were normalized using the method described above. The time-series data of each pixel were then processed using FFT, the PR0.25 during the 15 min of visual stimulation was calculated, and the magnitude of the PR0.25 was color-coded and mapped.

### Phase analysis

The phase shifts between the periodic oscillatory signals of the TexasRed and the dYFP signals, or between the TexasRed and the autofluorescence signals, were evaluated using the cross-correlation analysis. The lag time of the negative peak around 0 s in the cross-correlation was defined as the Δt. The temporal cycle of the horizontally oscillating visual stimuli was 4 s. A negative peak was usually found within the ±4 s range in the cross-correlation graph in almost all sessions. Data with no negative peak within this range were excluded from further analysis.

The phase shifts between the visual stimuli and the eye movement were calculated using the following method. The eye movement data during the first 5 min of each training was divided into segments of 4 s in duration, resulting in a series of 75 traces. These traces were averaged and fitted with a sinusoidal curve. The phase of the eye movement was calculated using this fitted curve to the averaged eye movement trace. The phase shifts between the visual stimuli and the TexasRed signal were analyzed similarly. However, as shown in *Figure 6F*, although the eye movement reliably followed the visual stimulation throughout the session, vasomotion apparently came in and out of the frequency locked to the visual stimulation. Therefore, PR0.25 was calculated for the TexasRed traces and only when the PR0.25 was higher than 3 that segment of the TexasRed trace was extracted. Using these selected segments of the TexasRed signals, the phase shift between the visual stimulation and the TexasRed signal was calculated.

### Statistical analysis

All statistical tests were done with Origin Pro 8.6 software (OriginLab, MA). Data are presented as means ± SEM, except in *Figure 7F*. The number of samples is indicated in the figure legend. Statistical analyses were performed with the paired *t*-test (two-sided), one-sample *t*-test, and Holm correction for multiple tests.

## Acknowledgements

We are thankful to all members of the Ko Matsui laboratory for their invaluable assistance at every stage of the experiments. This work was supported by JSPS Fellowships 22KJ0262 (to DS), Grant-in-Aid for

Early-Career Scientists 22K15218 (to YI), Grant-in-Aid for Transformative Research Areas (A) 'Glial Decoding': 20H05896 (to KM), 'Biology of Behavior Change': 23H04659 (to KM), Grant-in-Aid for Scientific Research on Innovative Areas 'Brain Information Dynamics': 18H05110, 20H05046 (to KM), Grant-in-Aid for Scientific Research (B): 19H03338, 22H02713 (to KM), Research Foundation for Opto-Science and Technology (to KM), Takeda Science Foundation (to KM), the NOVARTIS Foundation for the Promotion of Science (to KM), and the Uehara Memorial Foundation (to KM).

## Additional information

### Funding

| Funder | Grant reference number | Author |
| --- | --- | --- |
| Japan Society for the Promotion of Science | 22KJ0262 | Daichi Sasaki |
| Japan Society for the Promotion of Science | 22K15218 | Yoko Ikoma |
| Japan Society for the Promotion of Science | 20H05896 | Ko Matsui |
| Japan Society for the Promotion of Science | 23H04659 | Ko Matsui |
| Japan Society for the Promotion of Science | 18H05110 | Ko Matsui |
| Japan Society for the Promotion of Science | 20H05046 | Ko Matsui |
| Japan Society for the Promotion of Science | 19H03338 | Ko Matsui |
| Japan Society for the Promotion of Science | 22H02713 | Ko Matsui |
| Research Foundation for Opto-Science and Technology | | Ko Matsui |
| Takeda Science Foundation | | Ko Matsui |
| NOVARTIS Foundation | | Ko Matsui |
| Uehara Memorial Foundation | | Ko Matsui |

The funders had no role in study design, data collection and interpretation, or the decision to submit the work for publication.

### Author contributions

Daichi Sasaki, Conceptualization, Data curation, Formal analysis, Funding acquisition, Visualization, Methodology, Writing - original draft, Writing - review and editing; Ken Imai, Formal analysis, Validation, Methodology, Writing - review and editing; Yoko Ikoma, Supervision, Funding acquisition, Validation, Methodology, Writing - review and editing; Ko Matsui, Conceptualization, Resources, Supervision, Funding acquisition, Validation, Methodology, Writing - original draft, Project administration, Writing - review and editing

### Author ORCIDs

Ko Matsui ⃝iD http://orcid.org/0000-0003-1068-9705

### Ethics

This study was performed in strict accordance with the recommendations in the Regulations for Animal Experiments and Related Activities at Tohoku University. All of the animals were handled according to approved institutional animal care and use committee protocols of the Tohoku University

(2019LsA-017-10). All surgery was performed under anesthesia, and every effort was made to minimize suffering and to reduce the number of animals used.

Reviewer #2 (Public Review): https://doi.org/10.7554/eLife.93721.3.sa1

Reviewer #3 (Public Review): https://doi.org/10.7554/eLife.93721.3.sa2

Author response https://doi.org/10.7554/eLife.93721.3.sa3

# Additional files

## Supplementary files

• MDAR checklist

## Data availability

The datasets required to replicate the results and reproduce the figures in the manuscript are freely available in the project's Dryad Digital Repository (https://doi.org/10.5061/dryad.15dv41p4f).

The following dataset was generated:

| Author(s) | Year | Dataset title | Dataset URL | Database and Identifier |
|---|---|---|---|---|
| Sasaki D, Imai K, Ikoma Y, Matsui K | 2024 | Dataset for "Plastic vasomotion entrainment" in eLife | https://doi.org/10.5061/dryad.15dv41p4f | Dryad Digital Repository, 10.5061/dryad.15dv41p4f |

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
