## [Editor Report · eLife assessment]

This article presents **important** results indicating a plastic enhancement in the vasomotion response of pial cortical arterioles to external stimulation in awake mice using a wide range of external visual stimulation paradigms. The evidence for this interesting effect, with broad potential applications, is **solid**. These results are relevant for scientists and clinicians interested in the regulation of blood flow in the brain.

---

## [Referee Report · Reviewer #2 (Public Review)]

Sasaki et al. investigated methods to entrain vasomotion in awake wild-type mice across multiple regions of the brain using a horizontally oscillating visual pattern which induces an optokinetic response (HOKR) eye movement. They found that spontaneous vasomotion could be detected in individual vessels of their wild-type mice through either a thinned cranial window or intact skull preparation using a widefield macro-zoom microscope. They showed that low-resolution autofluorescence signals coming from the brain parenchyma could be used to capture vasomotion activity using a macro-zoom microscope or optical fibre, as this signal correlates well with the intensity profile of fluorescently-labelled single vessels. They show that vasomotion can also be entrained across the cortical surface using an oscillating visual stimulus with a range of parameters (with varying temporal frequencies, amplitudes, or spatial cycles), and that the amplitude spectrum of the detected vasomotion frequency increases with repeated training sessions. The authors include some control experiments to rule out fluorescence fluctuations being due to artifacts of eye movement or screen luminance and attempt to demonstrate some functional benefit of vasomotion entraining as HOKR performance improves after repeat training. These data add in an interesting way to the current knowledge base on vasomotion, as the authors demonstrate the ability to entrain vasomotion across multiple brain areas and show some functional significance to vasomotion with regards to information processing as HOKR task performance correlates well with vascular oscillation amplitudes.

---

## [Referee Report · Reviewer #3 (Public Review)]

Summary:

Here the authors show global synchronization of cerebral blood flow (CBF) induced by oscillating visual stimuli in the mouse brain. The study validates the use of endogenous autofluorescence to quantify the vessel "shadow" to assess the magnitude of frequency-locked cerebral blood flow changes. This approach enables straightforward estimation of artery diameter fluctuations in wild-type mice, employing either low magnification wide-field microscopy or deep-brain fibre photometry. For the visual stimuli, awake mice were exposed to vertically oscillating stripes at a low temporal frequency (0.25 Hz), resulting in oscillatory changes in artery diameter synchronized to the visual stimulation frequency. This phenomenon occurred not only in the primary visual cortex but also across a broad cortical and cerebellar surface. The induced CBF changes adapted to various stimulation parameters, and interestingly, repeated trials led to plastic entrainment. The authors control for different artefacts that may have confounded the measurements such as light contamination and eye movements but found no influence of these variables. The study also tested horizontally oscillating visual stimuli, which induce the horizontal optokinetic response (HOKR). The amplitude of eye movement, known to increase with repeated training sessions, showed a strong correlation with CBF entrainment magnitude in the cerebellar flocculus. The authors suggest that parallel plasticity in CBF and neuronal circuits is occurring. Overall, the study proposes that entrained "vasomotion" contributes to meeting the increased energy demand associated with coordinated neuronal activity and subsequent neuronal circuit reorganization.

Strengths:

-The paper describes a simple and useful method for tracking vasomotion in awake mice through an intact skull.

-The work controls for artefacts in their primary measurements.

-There are some interesting observations, including the nearly brain-wide synchronization of cerebral blood flow oscillations to visual stimuli and that this process only occurs after mice are trained in a visual task.

-This topic is interesting to many in the CBF, functional imaging, and dementia fields.

---

## [Author Response]

The following is the authors’ response to the original reviews.

**Recommendations for the authors:**

**Reviewer #1 (Recommendations For The Authors):**
Why does stimulation at 0.15 Hz show a third harmonic signal (Figure 5A) but 0.25 Hz does not show a second harmonic signal?

Second and third harmonic signals were sometimes observed in 0.15 Hz and also in 0.25 Hz and other frequency stimulations. The second harmonic signal is easier to understand as vasomotion may be reacting to both directions of oscillating stimuli. The reason for the emergence of the third harmonics was totally unknown. These harmonic signals were not always observed, and the magnitude of these signals was variable. The frequency-locked signal was robust, thus, in this manuscript, we decided to describe only this signal. These observations are mentioned in the revised manuscript (Results, page 9, paragraph 2).

References for the windows are missing. Closed craniotomy: (Morii, Ngai, and Winn 1986). Thinned skull: (Drew et al. 2010).

These references were incorporated into the revised manuscript.

An explanation of, or at least a discussion on, why a flavoprotein or other intrinsic signal from the parenchyma might follow vasomotion with high fidelity would be most helpful.

We spend a large part of the Results describing that any fluorescence signal from the brain parenchyma follows the vasomotion because the blood vessels largely lack fluorescence signals within the filter band that we observe. This is described as “shadow imaging”. What was rather puzzling was that flavoprotein or other intrinsic signals were phase-shifted in time. This suggests that these autofluorescence signals have an anti-phase “shadow imaging” component and another component that is phase-shifted in time. This is described in the manuscript as the following.

(Results, page 13, paragraph 2)

“Production and degradation of flavin and other metabolites may be induced by the fluctuation in the blood vessel diameter with a fixed delay time. The phase shift in the autofluorescence could be due to the additive effect of “shadow” imaging of the vessel and to the concentration fluctuation of the autofluorescent metabolite”

Glucose and oxygen are likely to be abundantly delivered during the vasodilation phase compared to the vasoconstriction phase of vasomotion. These molecules will trigger cell metabolism and endogenous fluorescent molecules such as NADH, NADPH, and FAD may increase or decrease with a certain delay, which is required for the chemical reactions to occur. Therefore, the concentration fluctuation of these metabolites could lag in time to the changes in the blood flow. These discussions are added in the revised manuscript (Discussions, page 19, paragraph 2).

**Reviewer #2 (Recommendations For The Authors):**
Minor corrections to the text and figures:(1) Figures 1 and 2- The single line slice basal and dilated traces are larger in Figure 2 (intact skull) than in Figure 1 (thinned skull)- have these been mixed up, as the authors state in the text that larger dilations are detected in the thinned skull preparation?

The example vessel described for the thinned skull (Figure 1) happened to be larger than that shown for the intact skull (Figure 2). We did not describe that larger dilations are observed in the thinned skull preparation. What was described was that the vessel profiles were shallower in the intact skull. This is because the presence of the intact skull blurs the fluorescence image.

(2) Figure 3- I think the lower panel of the amplitude spectrums from 3 individual animals included in D would benefit from being in its own panel within this Figure (i.e. E). The peak ratio is also used in this figure, but the equation to calculate this is not displayed until Figure 4.

We thank the reviewer for recommending making the figure more comprehensible. We have divided panel D into D and E and shifted the panel character accordingly. The manuscript text was also updated.

As the reviewer describes, the peak ratio of 0.25 Hz is used in Figure 3E (original). However, the equation to calculate this figure is described in the appropriate location within the main text of the manuscript (Results, page 10, paragraph 2) as well as in the figure legend.

(3) Figure 5- In the visual stimulation traces displayed in C you have included a 10-degree scale bar, which looks similar in amplitude to the trace but the text states these are 17-degree amplitude traces.

We thank the reviewer for noticing this mistake of labeling in the figure. We have corrected the error in the revised figure.

(4) Figure 6- For the Texas red fluorescence traces and image scales displayed in F, you have shown the responding traces on the right and non-responding on the left, but the figure legend states the amplitude is strong on the left and weak on the right.

We thank the reviewer for noticing the error in the figure legend text. We have corrected the error in the revised manuscript.

(5) Figure 6- It would be helpful for the reader if the r value was displayed on the graph in G.

We thank the reviewer for the suggestion. We have indicated the r value in Figure 6G as the reviewer recommended.

**Reviewer #3 (Recommendations For The Authors):**
MajorIt is unclear to me if the authors are studying vasomotion per se. Vasomotion is an intrinsic, natural rhythm of blood vessel diameter oscillation that is entrained by endogenous rhythmic neural activity. Importantly, if you take neural activity away, the blood vessel (with flow and pressure) should still be capable of oscillating due to an intrinsic mechanism within the vessel wall. In contrast, if one increases neural activity by way of sensory stimulation and blood flow increases, this is the basis of functional hyperemia. If one stimulates the brain over and over again at a particular frequency, it is expected that blood flow will increase whenever neural activity increases to the stimulus, up to a particular frequency until the blood vessel cannot physically track the stimulus fast enough. Functional hyperemia does not depend on an intrinsic oscillator mechanism. It occurs when the brain becomes active above endogenous resting activity due to sensory or motor activity.

We thank the reviewer for stressing the importance of the distinction between “vasomotion” and functional “hyperemia”.

We recognized that the terminology used in our paper was not explicitly explained. Traditionally, “vasomotion” is defined as the dilation and constriction of the blood vessels that occurs spontaneously at low frequencies in the 0.1 Hz range without any apparent external stimuli. Sensory-induced changes in the blood flow are usually called “hyperemia”. However, in our paper, we used the term, vasomotion, literally, to indicate both forms of “vascular” “motion”. Therefore, the traditional vasomotion was called “spontaneous vasomotion” and the hyperemia, with both vasoconstriction and vasodilation, induced with slow oscillating visual stimuli was called “visually induced vasomotion”. This distinction in the terminology is now explicitly introduced in the revised manuscript (Introduction, page 3, paragraph 2-3; page 4, paragraph 1-2).

Using our newly devised methods, we show the presence of “spontaneous vasomotion”. However, this spontaneous vasomotion was often fragmented and did not last long at a specific frequency. With visual stimuli that slowly oscillated at temporal frequencies close to the frequency of spontaneous vasomotion, oscillating hyperemia, or “visually induced vasomotion” was observed. Importantly, this visually induced vasomotion is not observed in novice animals. Therefore, the visually induced vasomotion is not a simple sensory reaction of the vascular in response to neuronal activity in the primary visual cortex. We also do not know how the synchronized vasomotion can spread throughout the whole brain. Where the plasticity for vasomotion entrainment occurs is also unknown. How much of the visually induced vasomotion relies on the mechanisms of intrinsic spontaneous vasomotion is also undetermined. Discussion about the future directions of understanding the mechanisms of visually induced vasomotion and entrainment is described in better detail in the revised manuscript (Discussions, page 19, paragraph 1).

To me, one would need to silence the naturally occurring vasomotion to study it. As soon as one activates the brain with an external stimulus, functional hyperemia is being studied. One idea that would be interesting to look at is whether a single or perhaps a double stimulus, in an untrained vs trained mouse, shows vasodilation that occurs across the cortex and in the cerebellum. In other words, is there something special about repeating the signal over and over again that results in brain-wide synchronization, or does a single or double oscillation of the same frequency (0.25Hz) also transiently synchronize the brain? My guess is that a short stimulus would give you the same thing (especially in a trained mouse) and that there is nothing special about oscillating the signal over and over again (except for the learning component).

We thank the reviewer for the ideas of new experiments to understand whether the visually induced vasomotion shares the same mechanisms for creating spontaneous vasomotion or not.

We would like to emphasize again that the visually induced vasomotion is not observed in the Novice animals. Therefore, the visually induced vasomotion is not a simple sensory reaction of the vascular in response to the visual stimuli. Entrainment with repeated presentation of visual stimuli is required for this global synchronization phenomenon to occur.

We would also like to emphasize that, even in Expert animals, the visually induced vasomotion that is frequency-locked to the presented stimulus does not always occur immediately. As shown in Figure 3D lower panel (Figure 3E in the revised figure), the vasomotion did not always immediately frequency-lock. The vasomotion was also not always stable throughout the 15 min of visual stimulation presentation. These characteristics are emphasized in the revised manuscript (Results, page 10, paragraph 1).

Therefore, we would assume that a single or double frequency of the visual stimulation would not always be sufficient to transiently frequency-lock the visually induced vasomotion.

An alternative idea is to test frequencies lower than vasomotion. Vasomotion typically oscillates around a wide range of very low frequencies averaging around 0.1Hz, yet here the authors entrain blood vessel oscillations towards the top end of vasomotion, at 0.25Hz. What would happen if the authors tried synchronizing brain activity with 0.025Hz? Would the natural vasomotion frequency still be there, or would it be gone, dominated by the 0.025Hz entrainment?

We would assume that visually induced vasomotion will not be induced with 0.025 Hz visual stimuli. This is too slow to induce smooth pursuit of the visual stimuli with eye movement. We show that, even if smooth eye pursuit occurs, the visually induced vasomotion may or may not occur (Figure 6F). However, visually induced vasomotion does not largely occur without eye movement. Therefore, the proposed experiment by the reviewer is likely not doable.

Finally, perhaps the authors can see if there is a long-lasting change in natural vasomotion occurring after the animal has been trained to 0.25Hz. For example, is there greater power in the endogenous fluctuation at either 0.25Hz (or perhaps 0.1Hz) with no visual stimulation given but after the animal has been trained? These ideas would be interesting to test and could help clarify whether this is plasticity in functional hyperemia or plasticity in vasomotion.

It should also be mentioned that the frequency-locked vasomotion quickly dissipates as soon as the visual stimulation is halted (Figure 3D upper panel, middle). However, we agree with the reviewer that it would be interesting to see whether the fragmentation of the spontaneous vasomotion is observed less in the Trained or Expert mice compared to the Novice mice, to understand whether the entrainment effect would propagate to the properties of the spontaneous vasomotion.

This issue I have raised is not a fundamental flaw in the paper, it pertains more to the wording, phrasing, and pitch of the paper i.e. is this really entrained and plastic vasomotion? I am skeptical. Nevertheless, I think the authors should try some of these suggestions to better characterize this effect.

We agree that the phrasing used in the original manuscript was rather confusing, as “vasomotion” normally refers to spontaneous vascular movement. However, functional “hyperemia” may not adequately express the phenomenon that we observe either. The phenomenon that we observe is slowly oscillating vasodilation and vasoconstriction that is induced with visual stimuli with a temporal frequency similar to the spontaneously occurring “vasomotion”. This phenomenon is not a direct hyperemia response to the visual stimuli as it requires entrainment and it spreads globally throughout the whole brain. We revised our manuscript to define the terminology that we use.

An important question is if neural activity is entraining the CBF responses. The authors should do one experiment in a pan-neural GCaMP line to test if neural activity in the visual cortex (and other areas captured in the widefield microscope) shows a progressive and gradual synchronization (or not) to the vasomotion responses with training. It is possible to do this through a thinned skull window. This important to know if/how synchronized population neural activity scales with training. Perhaps they will not correlate and there is something more subtle going on.

In our paper, we mainly studied visually induced vasomotion (or visual stimulus-triggered vasomotion). Therefore, visual stimulation must first activate the neurons and, through neurovascular coupling, the initial drive for vasomotion is likely triggered. However, visually induced vasomotion is not observed in novice animals. Therefore, the visually induced vasomotion is not a simple sensory reaction of the vascular in response to neuronal activity in the primary visual cortex.

An important point that should be pointed out is that the neuronal visual response in the primary visual cortex could potentially decrease with repeated visual stimulation presentation as the adaptive movement of the eye should decrease the retinal slip. With repeated training sessions, a more static projection of the presented image will likely be shown to the retina. The neurovascular coupling could be enhanced with increased responsiveness of the vascules and vascular-to-vascular coupling could also be potentiated. This argument is now incorporated in the revised manuscript (Discussions, page 19, paragraph 1).

We agree with the reviewer that, to identify the extent of the neuronal contribution to the vasomotion triggering, whole brain synchronization, and vasomotion entrainment, simultaneous neuronal calcium imaging would be ideal. However, due to the fact that fluorescent Ca2+ indicators expressed in neurons would also be distorted by the “shadow” effect from the vasomotion, exquisite imaging techniques would be required. We recognize this “shadow” effect and we are currently developing methods to take out the “shadow” effect and the intracellular pH fluctuation effect from the fluorescence traces.

The authors nicely show that plasticity in vasomotion coincides with the mouse learning the HOKR task and that as eye movement tracks the stimulus, CBF gets entrained. However, there could also be a stress effect going on in the early trials, and as the mouse gets used to the procedure and stress comes down, the vasomotion entrainment can be seen. It could be the case that the vasomotion process is there on the first trial, but masked by stress-induced effects on neural and/or vascular activity. I did not see anything in the methods about how the mouse was habituated to head restraint. Was the first visual stim trial the first time the mouse was head restrained? If so, there could be a strong stress effect. The authors should address this either by clarifying that habituation to head restraint was done, or by doing a control experiment where each animal receives at least 1week of progressive and gradual head restraint before doing the same HOKR experiment using multiple trials.

We agree with the reviewer that stress could well affect spontaneous vasomotion as well as visually induced vasomotion (or visual stimulus-triggered vasomotion). As the reviewer suggested, we could have compared the habituated and non-habituated mice to the initial visually induced vasomotion response. In addition, whether the experimentally induced increase in stress would interfere with the vasomotion or not could also be studied. With the TexasRed experiments, we observed that tail-vein injection stress appeared to interfere with the HOKR learning process. In the experiments presented in Fig. 3, TexasRed was injected before session 1. Vasomotion entrainment likely progressed with sessions 2 and 3 training. Before session 4, TexasRed was injected again to visualize the vasomotion. The vasomotion was clearly observed in session 4, indicating that the stress induced by tail-vein injection could not interfere with the generation of visually induced vasomotion. This argument is included in the revised manuscript (Discussions, page 20, paragraph 2).

MinorThe first sentence of the introduction requires citations. It is also a somewhat irrelevant comparison to make.

Necessary citation was made in the revised manuscript, as the reviewer suggested. We think that describing how the energy is distributed in the brain would provide one of the most important breakthroughs to the understanding of how efficient information processing in the brain works. Therefore, we would like to keep this introduction.

The third and fourth sentence of the introduction equates vasodilation/vasoconstriction with vasomotion and it is not this simple. Vasomotion is a specific physiological process involving rhythmic changes to artery diameter. Also, the frequency of these slow oscillations needs to be stated. The authors only say they are slower than 10Hz.

The definition of spontaneous vasomotion with indication of typical temporal frequency is described in the revised manuscript, as the reviewer suggested.

More than half of the introduction is describing the paper itself, rather than setting the stage for the findings. The authors need a more thorough account of what is known and what is not known in this area. Some of this information is in the discussion, which should be moved up to the intro.

We have revised the introduction to include the definition of spontaneous vasomotion and visually induced vasomotion or functional hyperemia, as the reviewer suggested.

In the first paragraph of the results section, the authors should state in what way the mice are awake. Are they freely mobile? Are they head-restrained? Are they resting or moving or doing both at different times? This is clarified later but it should come up front as someone reads through the paper.

As the reviewer suggested, we clarified that the experiments were done in awake and head-restrained mice within the first paragraph for the Results section.

The authors say "As shown later, blood vessels on the surface...". There is no need to say "as shown later".

This is deleted as the reviewer suggested.

The use of "full width at 10% maximum" of the Texas red intensity for the diameter measure is a little odd, as it may actually overestimate the diameter, but I see what the authors were trying to do. A full-width half max is standard here and that is likely more appropriate. Also, the line profiles of intensity are not raw data. The authors say the trace is strongly filtered/smoothed. If so, this creates a somewhat artificial platform to make the diameter measurement. The authors should show raw data from a single experiment and make the measurement from that. The raw line profile should look almost square, where a full-width half-max would work well.

Contrary to what the reviewer observed, the raw line profile was not almost square. Even if there were almost no blur in the XY dimension in the optical imaging system, one would not expect to see a square line profile, as the thickness of the vessel increases in the Z dimension towards the center, as this is not a confocal or two-photon microscope image, and an ideal optical section was not created. Therefore, the full-width half-maximum value would definitely be an underestimate of the actual vessel diameter. It may be possible to equate an ideal value for cutoff if we have the 3D point spread function of the imaging. 10% is an arbitrary number but we think 10% is the minimum intensity that we can distinguish from the background intensity fluctuations. We did not attempt to derive the “true” diameter of the vessel and full-width at 10% maximum is just an index of the actual diameter. In most of the manuscript, we only deal with the change of the vessel diameter relative to the basal diameter, therefore, we considered that careful derivation of the absolute diameter estimate is not necessary. This argument is detailed in the Materials and Methods section in the revised manuscript (page 31, paragraph 2).

The raw line profile before filtering is shown overlaid in Figure 1C, as the reviewer suggested.

In Figures 1 and 2, state/label what brain region this is.

The blood vessels between the bregma and lambda on the cortex were observed and described in Figures 1 and 2. This is described in the revised manuscript, as the reviewer suggested.

Can the authors also show what a vein or venule looks like using their quantification method in Figures 1 and 2? This would be a helpful comparison to a static vein.

The methods shown in Figures 1 and 2 would not allow us to distinguish between vein and venule in our study. Methods that allow quantification of the relative blood vessel diameter fluctuation due to spontaneous or visually induced vasomotion activities are shown in Figures 1 and 2. Later in the manuscript, the whole intensity fluctuation of TexasRed or autofluorescence in the brain parenchyma is studied, and in this case, no distinction between vein and venules could be made.

Statements such as this are not necessary: "Later in the manuscript, we will be dealing with vasomotion dynamics observed with the optical fiber photometry methods, in which the blood vessel type under the detection of the fiber could not be identified". Simply talk about this data when you get to it.

We have deleted this statement in this part of the manuscript, as the reviewer suggested.

Same as this, please consider deleting: "Spontaneous vasomotion dynamic differences between different classes of blood vessels would be of interest to study using a more sophisticated in vivo two-photon microscope which we do not own." Just describe the data you have from the methods you have. There is no need to lament.

We deleted this sentence, as the reviewer suggested.

Figure 3 D the light blue boxes showing the time period of visual stimulation physically overlay with the frequency-time spectrograms. They should not overlay with this graph because it makes them more light blue, distorting the figure which also uses light blue in the heat map.

Figure 3D was modified, as the reviewer suggested.

The authors say: "The reason why the vasomotion detected in our system through the intact skull in awake in vivo mice was less periodic was unknown." Yes, but you are imaging an awake mouse. Many spontaneous behaviours such as whisking, grooming, twitching, and struggling will manifest as increased artery diameter. These will be functional hyperemia occurring events on top of rhythmic vasomotion. This can be briefly discussed.

As the reviewer comments, the vasomotion detected in awake mice was likely to be less periodic because the spontaneous animal behavior induces functional hyperemia and interrupts spontaneous vasomotion. This interpretation was included in the revised manuscript (Results, page 8, paragraph 1).

The authors say "extremely tuned" on page 8. They should not use words like "extremely". Perhaps say "more strongly tuned" or equivalent.

We have changed “extremely” to “more strongly”, as the reviewer suggested.

The authors say "First, the Texas Red fluorescence images were Gaussian filtered in the spatial XY dimension to take out the random noise presumably created within the imaging system." It is inadvisable to alter the raw data in this way unless there is a sound reason to do so. If there is random noise this should not affect the Fast Fourier Transform analysis. If there is regular noise caused by instrumentation artefact, which is picked up by the analysis then perhaps this could be filtered out. A static Texas red sample in a vial can be used to determine if there is artefactual noise.

We mainly used the Gaussian filter for better presentation of the imaged data. The TexasRed fluorescence was low in intensity and the acquired images were Gaussian filtered in the spatial XY dimesion to reduce the pixelated noise at the expense of spatial resolution reduction. This filter should not affect the temporal frequency of the observed vasomotion. This is now more clearly indicated in the revised manuscript (Results, page 10, paragraph 2).

There are endogenous fluorescent molecules in cell metabolism that change dynamically to neural activity: NADH, NADPH, and FAD. These are almost certainly a fraction of the auto-fluorescent signal the authors are measuring and it would be expected to see small fluctuations in these metabolites with neural activity. Perhaps this can be discussed, and the authors can likely argue that metabolic signals are much smaller than the change caused by vasodilation.

We found that the autofluorescence signal was phase-shifted in time relative to the vasomotion, which was visualized with TexasRed. This suggests that these autofluorescence signals have an anti-phase “shadow imaging” component and another component that is phase-shifted in time. Glucose and oxygen are likely to be abundantly delivered during the vasodilation phase compared to the vasoconstriction phase of vasomotion. These molecules will trigger cell metabolism and endogenous fluorescent molecules such as NADH, NADPH, and FAD may increase or decrease with a certain delay, which is required for the chemical reactions to occur. Therefore, the concentration fluctuation of these metabolites could lag in time to the changes in the blood flow. It is also expected that these metabolites may fluctuate according to the neuronal activity that triggers visually induced vasomotion or functional hyperemia. These discussions are added in the revised manuscript (Discussions, page 19, paragraph 2).

The authors say "however, we found that, if Texas Red had to be injected before every training session, the mouse did not learn very well." This is interesting. Why do the authors suppose this was the case? Stress from the injection? Or perhaps some deleterious effect on blood vessel function caused by the dye itself? Either way, I think this honest statement should remain. Others need to know about it.

We think that the stress from the injection interferes with the HOKR learning. However, as shown, TexasRed injection after the mouse had learned did not interfere with the eye movement or with the visually induced vasomotion. We do not know whether the injection stress directly interferes with the blood vessel function and affects the plastic vasomotion entrainment. These arguments are now described in the revised manuscript (Discussions, page 20, paragraph 2). The statement above remains as is, as the reviewer suggested.

YCnano50 is a calcium sensor and not really appropriate for the use employed by the authors. They are exciting YFP at 505nm but unless the authors are using a laser line, there is some bandwidth of excitation light that is likely exciting the CFP too which still absorbs light up to ~490nm. Here, calcium signalling may affect the YFP signal. This can be discussed.

Multiband-pass filter (Chroma 69008x with the relevant band of 503 nm / 19.5 nm (FWHM)) was used for direct excitation of YFP. Negligible light is passed below 490 nm. CFP excitation above 490 nm is assumed to be negligible and usually not defined in literature. We assume that with our optical system, fluorescence by direct YFP excitation dominates the effect from the minor CFP excitation effect. We explicitly describe this in the revised manuscript (Materials and Methods, page 28, paragraph 2).

The discussion is interesting but does not actually discuss much of the data or measurements in the paper. Most of the discussion reads more like a topical review, rather than a critical analysis of the effects/measurements and why the authors' interpretations are likely correct. This can be improved.

As the reviewer suggests, we have improved the discussion by starting with the summary of the results (Discussion, page 19, paragraph 1). We also included the possibility of stress affecting visually induced vasomotion (Discussion, page 20, paragraph 2).